# Enhancer architecture sensitizes cell specific responses to *Notch* gene dose via a bind and discard mechanism

Yi Kuang[1†], Ohad Golan[2†], Kristina Preusse[3], Brittany Cain[4], Collin J Christensen[5], Joseph Salomone[1,6], Ian Campbell[4], FearGod V Okwubido-Williams[4], Matthew R Hass[3], Zhenyu Yuan[5], Nathanel Eafergan[2], Kenneth H Moberg[7], Rhett A Kovall[5], Raphael Kopan[3,8]*, David Sprinzak[2]*, Brian Gebelein[3,8]*

[1]Graduate Program in Molecular and Developmental Biology, Cincinnati Children's Hospital Research Foundation, Cincinnati, United States; [2]School of Neurobiology, Biochemistry and Biophysics, George S. Wise Faculty of Life Science, Tel Aviv University, Tel Aviv, Israel; [3]Division of Developmental Biology, Cincinnati Children's Hospital, Cincinnati, United States; [4]Department of Biomedical Engineering, University of Cincinnati, Cincinnati, United States; [5]Department of Molecular Genetics, Biochemistry and Microbiology, University of Cincinnati College of Medicine, Cincinnati, United States; [6]Medical-Scientist Training Program, University of Cincinnati College of Medicine, Cincinnati, United States; [7]Department of Cell Biology, Emory University and Emory University School of Medicine, Atlanta, United States; [8]Department of Pediatrics, University of Cincinnati College of Medicine, Cincinnati, United States

*For correspondence:
raphael.kopan@cchmc.org (RK);
davidsp@post.tau.ac.il (DS);
brian.gebelein@cchmc.org (BG)

[†]These authors contributed equally to this work

Competing interests: The authors declare that no competing interests exist.

**Abstract** Notch pathway haploinsufficiency can cause severe developmental syndromes with highly variable penetrance. Currently, we have a limited mechanistic understanding of phenotype variability due to gene dosage. Here, we unexpectedly found that inserting an enhancer containing pioneer transcription factor sites coupled to Notch dimer sites can induce a subset of *Notch* haploinsufficiency phenotypes in *Drosophila* with wild type *Notch* gene dose. Using *Drosophila* genetics, we show that this enhancer induces Notch phenotypes in a Cdk8-dependent, transcription-independent manner. We further combined mathematical modeling with quantitative trait and expression analysis to build a model that describes how changes in Notch signal production versus degradation differentially impact cellular outcomes that require long versus short signal duration. Altogether, these findings support a 'bind and discard' mechanism in which enhancers with specific binding sites promote rapid Cdk8-dependent Notch turnover, and thereby reduce Notch-dependent transcription at other loci and sensitize tissues to gene dose based upon signal duration.

## Introduction

Haploinsufficiency, or the inability to complete a cellular process with one functional allele of a given gene, manifests in tissue and organ defects with variable penetrance and severity (*Wilkie, 1994*). For example, *Notch* (*N*) haploinsufficiency, which was discovered in *Drosophila*, causes a variety of tissue-specific defects including wing notching and extra sensory bristle formation that can vary greatly in penetrance and expressivity (*Mohr, 1919*). Notch pathway haploinsufficiency was subsequently observed in mammals, as *Notch1* heterozygous mice have heart valve and endothelium defects (*Nigam and Srivastava, 2009*), whereas *Notch2* heterozygotes have defects in bone, kidney

and marginal zone B cells (*Isidor et al., 2011*; *Simpson et al., 2011*; *Witt et al., 2003*). A single allele of *NOTCH2* or the *JAG1* ligand can also cause pathological phenotypes in humans, as heterozygosity of either gene can result in a variably penetrant developmental syndrome known as Alagille (*McDaniell et al., 2006*; *Li et al., 1997*; *Oda et al., 1997*). Thus, *Notch* gene dose sensitivity has been observed in a variety of Notch-dependent tissues in both humans and animals. Unfortunately, we currently lack a mechanistic understanding of what causes some tissues to be highly sensitive to *Notch* gene dose and what factors impact the variable penetrance and severity of *Notch* haploinsufficiency phenotypes.

Molecularly, Notch signaling is initiated by ligand-induced proteolysis of the Notch receptor to release the Notch intracellular domain (NICD) from the membrane (*Kovall et al., 2017*; *Bray, 2016*). NICD subsequently transits into the nucleus, binds to the <u>C</u>bf1/<u>Su</u>(H)/<u>L</u>ag1 (CSL) transcription factor (TF) and the adaptor protein Mastermind (Mam), and induces gene expression via two types of DNA binding sites: independent CSL sites that bind monomeric NICD/CSL/Mam (NCM) complexes, and <u>Su</u>(H) <u>p</u>aired <u>s</u>ites (SPS) that are oriented in a head-to-head manner to promote cooperative binding between two NCM complexes (*Kovall et al., 2017*; *Bray, 2016*). Once bound to an enhancer, the NCM complex activates transcription of associated genes via the P300 co-activator. Thus, the production of NICD is converted into changes in gene expression that ultimately regulate cellular processes during development.

Haploinsufficiency of Notch receptor and ligand encoding genes suggests that decreased gene dosage results in a sufficiently large decrease in NICD production to cause phenotypes in a subset of tissues. There is also growing evidence that genetic changes that reduce NICD degradation can alter signal strength with pathological consequences in specific cell types. In the mammalian blood system, for example, *Notch1* mutations that remove an NICD degron sequence have been associated with increased NICD levels and the development of T-cell Acute Lymphoblastic Leukemia (T-ALL) in mice and humans (*O'Neil et al., 2006*; *Weng et al., 2004*). Intriguingly, NICD turnover via this degron sequence has been directly linked to transcription activation, as the Mam protein interacts with the Cdk8 kinase module (CKM) of the Mediator complex, which can phosphorylate NICD to promote its ubiquitylation by the Fbxw7 E3-ligase and degradation by the proteasome (*Fryer et al., 2004*; *Fryer et al., 2002*). Accordingly, gene mutations that lower CKM activity have also been associated with increased NICD levels and T-ALL initiation and progression (*Li et al., 2014*). Thus, perturbations in mechanisms that regulate either NICD production or degradation can induce cell and/or tissue specific phenotypes.

In this study, we use *Drosophila* genetics, quantitative trait and expression analysis, and mathematical modeling to unravel a unique regulatory mechanism that impacts Notch signal strength in a tissue-specific manner. First, we unexpectedly found that an enhancer containing as few as 12 Notch dimer binding sites can induce tissue-specific phenotypes via a CKM-dependent mechanism that can be uncoupled from transcription activation. Second, based on our quantitative analysis and mathematical modeling, we show how changes in NICD degradation rates are predicted to preferentially impact long duration Notch-dependent processes, whereas genetic changes in NICD production rates (i.e. *Notch* haploinsufficiency) affect both short and long duration processes. Collectively, these findings provide new insights into how distinct Notch-dependent cellular processes can be differentially impacted by both enhancer architecture and signal duration to induce tissue-specific Notch defects within a complex animal.

## Results

### Enhancers with specific TF binding sites can induce a tissue-specific *Notch* phenotype

To better understand transcriptional responses to *Notch* signals in *Drosophila*, we designed synthetic enhancers with comparable numbers of either CSL monomer or SPS dimer sites (*Figure 1A*; note, 1xSPS has the same number of sites as 2xCSL) (*Arnett et al., 2010*; *Nam et al., 2007*; *Bailey and Posakony, 1995*). We first tested the synthetic 1xSPS and 2xCSL sites for their ability to bind Notch/CSL/Mastermind (NCM) complexes in electromobility shift assays (EMSAs) using purified mouse (RBPJ, N1ICD1, and MAML1) and *Drosophila* (Su(H), NICD, and Mam) proteins. For this experiment, an equal amount of differentially labeled 2xCSL (IRdye-700, pseudo-colored magenta)

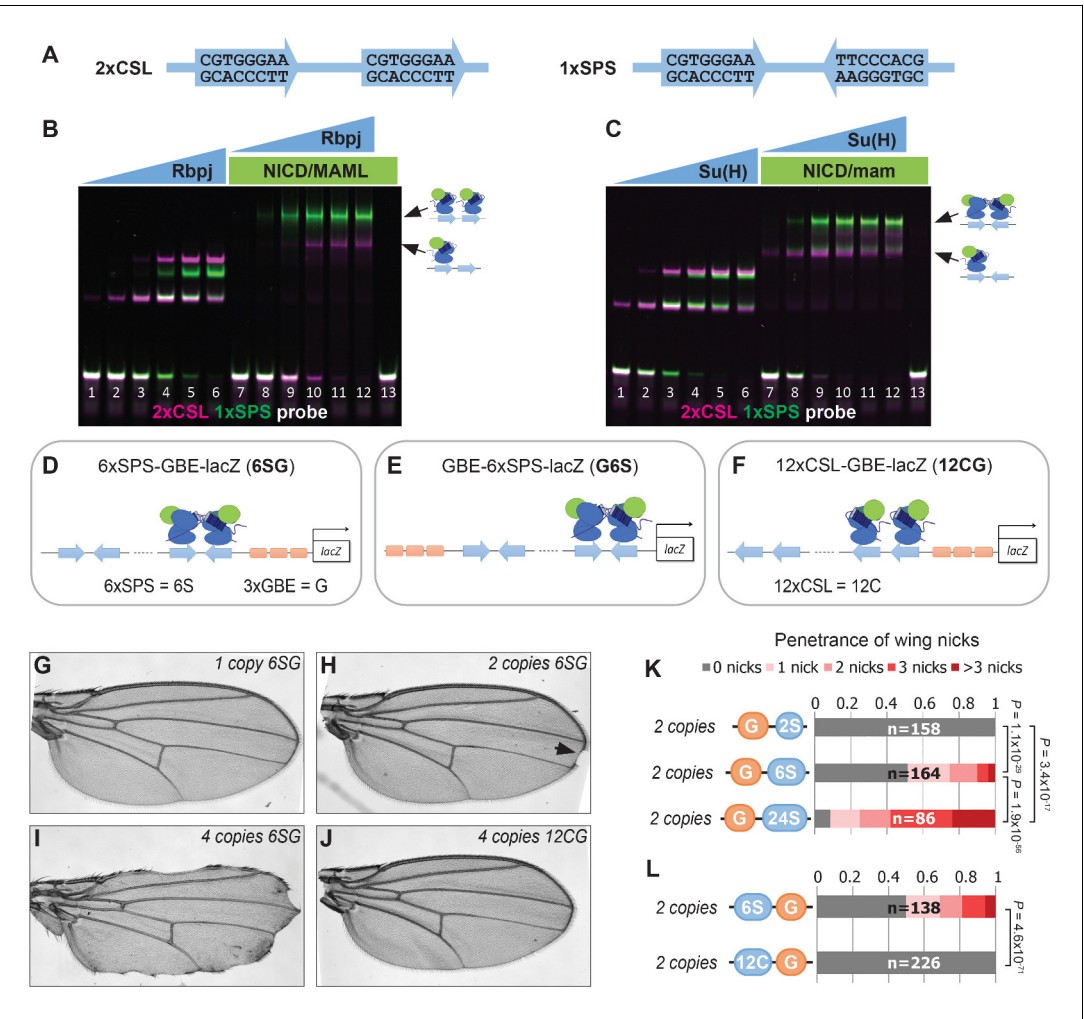

**Figure 1.** Synthetic Notch enhancers induce a *Drosophila* notched wing phenotype. (**A**) Sequences of 2xCSL and 1xSPS used for performing EMSAs and generating reporters. (**B–C**) Electromobility shift assays (EMSAs) using purified mammalian NCM proteins (**B**) and fly NCM proteins (**C**) with both 2xCSL (magenta) and 1xSPS (green) probes. Arrows highlight bands consistent with one vs two NCM complexes on DNA. RBPJ/Su(H) concentration increases from 10 to 320 nM and 1 μM NICD/Mastermind was used in indicated lanes. Note, we separated the 2xCSL and 1xSPS probe signals and show the EMSA data in grayscale in *Figure 1—figure supplement 1*. (**D–F**) Schematics of *6xSPS-3xGBE-lacZ* (*6SG*), *3xGBE-6xSPS-lacZ* (*G6S*) and *12xCSL-3xGBE-lacZ* (*12CG*). (**G–J**) Wings from flies with one copy, two copies, or 4 copies of *6SG-lacZ*, or 4 copies of *12CG-lacZ*. Arrowhead denotes a notch. (**K–L**) Quantified wing notching in flies with indicated transgenes. Proportional odds model tested penetrance and severity differences between *G6S-lacZ* and *G24S-lacZ*. Two-sided Fisher's exact test assessed the penetrance of other genotypes.

The online version of this article includes the following figure supplement(s) for figure 1:

**Figure supplement 1.** Individual channels of EMSAs.

**Figure supplement 2.** GBE and SPS sites are both required to induce the formation of wing nicks.

and 1xSPS (IRdye-800, pseudo-colored green) probe was added into the same binding reaction containing different concentrations of the RBPJ/Su(H) TF with and without the species matched NICD/Mastermind co-activators (*Figure 1B–C* and *Figure 1—figure supplement 1*). In the absence of co-activators, RBPJ and Su(H) bound each probe additively with both the mouse and fly proteins showing a slightly higher affinity to 2xCSL than 1xSPS (note, the increased unbound 1xSPS probe (green) in lane 4 of *Figure 1B* and lane 3 of *Figure 1C* relative to unbound 2xCSL probe (magenta)). Including the NICD/Mam co-activators didn't change the additive binding behavior of RBPJ and Su(H) to 2xCSL. By contrast, both the mouse and fly NCM complex preferentially filled both sites of the

1xSPS probe relative to 2xCSL (*Figure 1B–C*). Hence, these results demonstrate that while both the 2xCSL and 1xSPS synthetic sequences bind NCM complexes, only the 1xSPS sites mediate cooperative NCM complex binding.

We next generated transgenic fly lines containing reporter genes with either CSL or SPS binding sites. Since prior studies found that including sites for the Grainyhead (Grh) pioneer TF enhanced Notch reporter activity (*Furriols and Bray, 2001*) and induced chromatin opening (*Jacobs et al., 2018*), we generated fly lines containing CSL and SPS reporters with (*Figure 1D–F*) and without (*Figure 1—figure supplement 2B–C*) three copies of a Grh binding element (3xGBE). Surprisingly, flies homozygous for *6xSPS-3xGBE-lacZ* (*6SG-lacZ*) developed a notched-wing phenotype that mimics a classic *Notch* haploinsufficiency (*Figure 1H*). In contrast, flies homozygous for *3xGBE* alone (*G-lacZ*), *6xSPS* alone (*6S-lacZ*), or mutated SPSs (*6SmutG-lacZ*) (*Tun et al., 1994*) inserted in the same locus were indistinguishable from wild type (*Figure 1—figure supplement 2A–C*). To define the *SPS-GBE* binding site features that contribute to wing notching, we tested additional fly lines and found that: i) The *6SG-lacZ* caused notched wings when inserted in another locus and regardless of the order of GBE and SPS, although with differences in penetrance and severity (*Figure 1—figure supplement 2D–H*); ii) The penetrance and severity of wing notching increased as a function of both transgene and SPS numbers (*Figure 1G–I and K*); and iii) Flies with an equal number of Notch monomer (CSL) sites next to 3xGBE did not develop notched wings (*Figure 1J and L*). In total, these findings show that adding as few as 12 GBE-associated SPSs into the genome is sufficient to induce a *Notch* haploinsufficiency phenotype in the wing.

To determine if the *6SG-lacZ* induced wing phenotype could be modified by genetic changes in *Notch* pathway components, we analyzed flies carrying different gene copy numbers of either *N* or the *Hairless* (*H*) co-repressor that antagonizes Notch-mediated gene activation (*Morel et al., 2001*; *Bang and Posakony, 1992*). We found that a single *6SG-lacZ* transgene greatly enhanced the penetrance and severity of wing notching in *N* heterozygotes compared to either genotype alone (*Figure 2A–C* and *Figure 2—figure supplement 1*). Moreover, adding two extra alleles of *N* (4N) or removing one allele of *H* significantly suppressed the notched wing phenotype induced by two copies of *6SG-lacZ* (*Figure 2D–H*). Thus, wing phenotypes induced by *6SG* can be enhanced or suppressed by changing the gene dose of *N* and *H*, respectively.

*N* and *H* haploinsufficiency also cause defects in macrochaetae bristle patterning (*Bang et al., 1991*; *Shellenbarger and Mohler, 1978*) and wing vein development (*de Celis and García-Bellido, 1994*). Intriguingly, *6SG-lacZ* did not significantly impact macrochaetae formation in either wild type or sensitized $N^{+/-}$ and $H^{+/-}$ backgrounds (*Figure 2J–L*). However, while flies carrying two copies of the *6SG-lacZ* Notch-dimer reporter alone did not cause a noticeable wing vein phenotype, two copies of the *6SG-lacZ*, but not the *12CG-lacZ* Notch-monomer reporter, did significantly suppress the loss of L5 wing vein observed in $H^{+/-}$ animals (*Figure 2G–I*). Altogether, these data demonstrate that coupled *SPS-GBE* sites affect a subset of dose sensitive phenotypes with wing margin cells being the most sensitive.

## Cdk8 induces Notch turnover independent from transcription activation

Our findings raise two questions: How do the integrated *SPS-GBE* sites affect Notch activity, and why only in a subset of Notch-dependent processes? Prior studies have shown that Notch signal strength can be impacted by changes in either NICD production or degradation. Since NICD degradation in mammalian cells can be regulated by the Cdk8-Kinase module (CKM) that associates with the Mediator complex (*Li et al., 2014*; *Fryer et al., 2004*), we used genetics to assess the importance of the CKM in inducing wing phenotypes. To do so, we removed an allele of each gene of the Cdk8-kinase module (CKM; *cdk8* (*Loncle et al., 2007*), *cycC* (*Loncle et al., 2007*), *kto* (*Med12*) (*Treisman, 2001*), or *skd* (*Med13*) [*Treisman, 2001*]) or an allele of an E3-ligase that encodes the *Drosophila* homologue of *Fbxw7* (*archipelago, ago*) [*Moberg et al., 2001*]) and found that each significantly suppressed the penetrance and severity of 6SG-induced wing nicking (*Figure 3A* and *Figure 3—figure supplement 1A*). Notably, we observed that in *N* heterozygotes, removing an allele of *cycC*, *kto*, *skd* or *ago*, but not *cdk8*, also significantly suppressed wing notching (*Figure 3B* and *Figure 3—figure supplement 1B*). These data are consistent with Cdk8 phosphorylating NICD to promote its ubiquitylation and degradation (*Figure 3C*; *Li et al., 2014*; *Fryer et al., 2004*). Moreover, the smaller effect of *cdk8* gene dose compared to changes of the other CKM genes is consistent with studies showing that *cdk8* is not thought to be the limiting factor in the formation of this

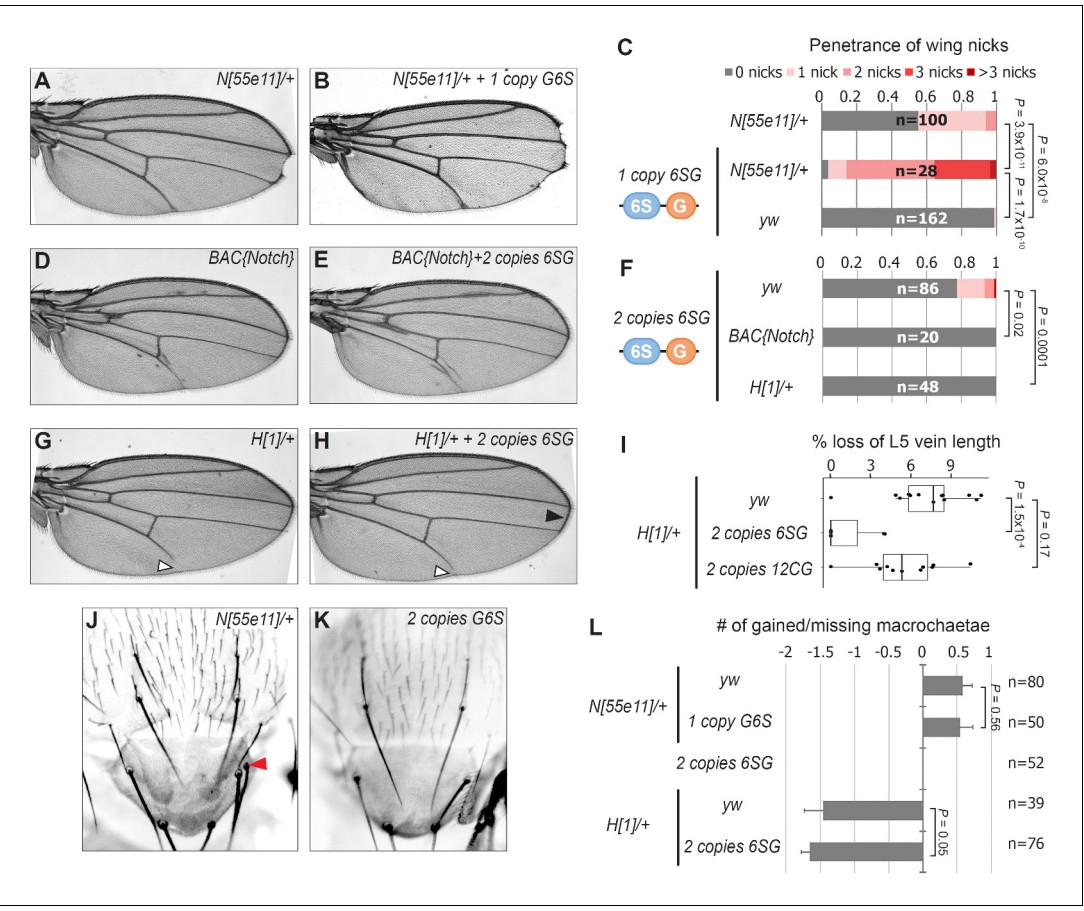

**Figure 2.** *SPS-GBE* reporters impact wing margin and vein development but not macrochaetae. (**A–B**) Wings from *Notch* heterozygotes ($N^{55e11/+}$) in the absence (**A**) and presence of *G6S-lacZ* (**B**). (**C**) Quantified wing notching in the indicated genotypes. Proportional odds model tested for penetrance/severity differences. (**D–E**) Wings from flies containing two extra *N* alleles (BAC{Notch-GFP-FLAG}) in the absence (**D**) and presence (**E**) of *6SG-lacZ*. (**F**) Quantified wing notching in flies with indicated genotypes. Two-sided Fisher's exact test. (**G–H**) Wings from $H^{1/+}$ flies in the absence (**G**) and presence (**H**) of *6SG-lacZ*. Solid arrowhead highlights loss (**G**) and rescue (**H**) of L5 wing vein. Open arrowhead points to rescued *6SG*-induced wing notching phenotype in $H^{1/+}$ flies. (**I**) Quantification of loss of L5 vein in flies with indicated genotypes. Each dot represents a measurement from an individual wing. Two-sided Student's t-test. In box plots, the line represents median, the box shows interquartile range, and whiskers represent the 1.5 times interquartile range. (**J–K**) Notum images from a $N^{55e11/+}$ (**J**) and *G6S-lacZ* (**K**) fly. Arrowhead denotes extra macrochaetae in $N^{55e11/+}$. (**L**) Quantification of gained/lost dorsalcentral and scutellar macrochaetae (wild type = 8) in indicated genotypes. Proportional odds model tested for statistical significance. Data are mean ±95% confidence interval.

The online version of this article includes the following figure supplement(s) for figure 2:

**Figure supplement 1.** *GBE-SPS* reporter enhances the wing notching phenotype in the $N^{1/+}$ background.

---

Mediator submodule (*Davis et al., 2013*; *Knuesel et al., 2009b*) and that enzymes are not typically gene dose sensitive (*Kondrashov and Koonin, 2004*). Hence, these findings support the model that lowering the dose of key CKM genes in *Drosophila* slows NICD turnover and thereby rescues the wing notching phenotype observed in *6SG-lacZ* and *N* heterozygotes.

The CKM has a complex relationship with promoter transcription (*Fant and Taatjes, 2019*). Some studies suggest interactions between the CKM and the core Mediator occludes RNA polymerase recruitment (*Knuesel et al., 2009a*) and/or decreases transcription (*Pelish et al., 2015*), whereas other studies suggest Cdk8 stimulates transcription (*Galbraith et al., 2013*; *Donner et al., 2010*). To test the role of the transgene promoter in causing wing nicks, we analyzed flies with promoter-containing (*3xGBE-24xSPS-GFP, G24S-GFP*) or promoter-less transgenes (*3xGBE-24xSPS, G24S*)

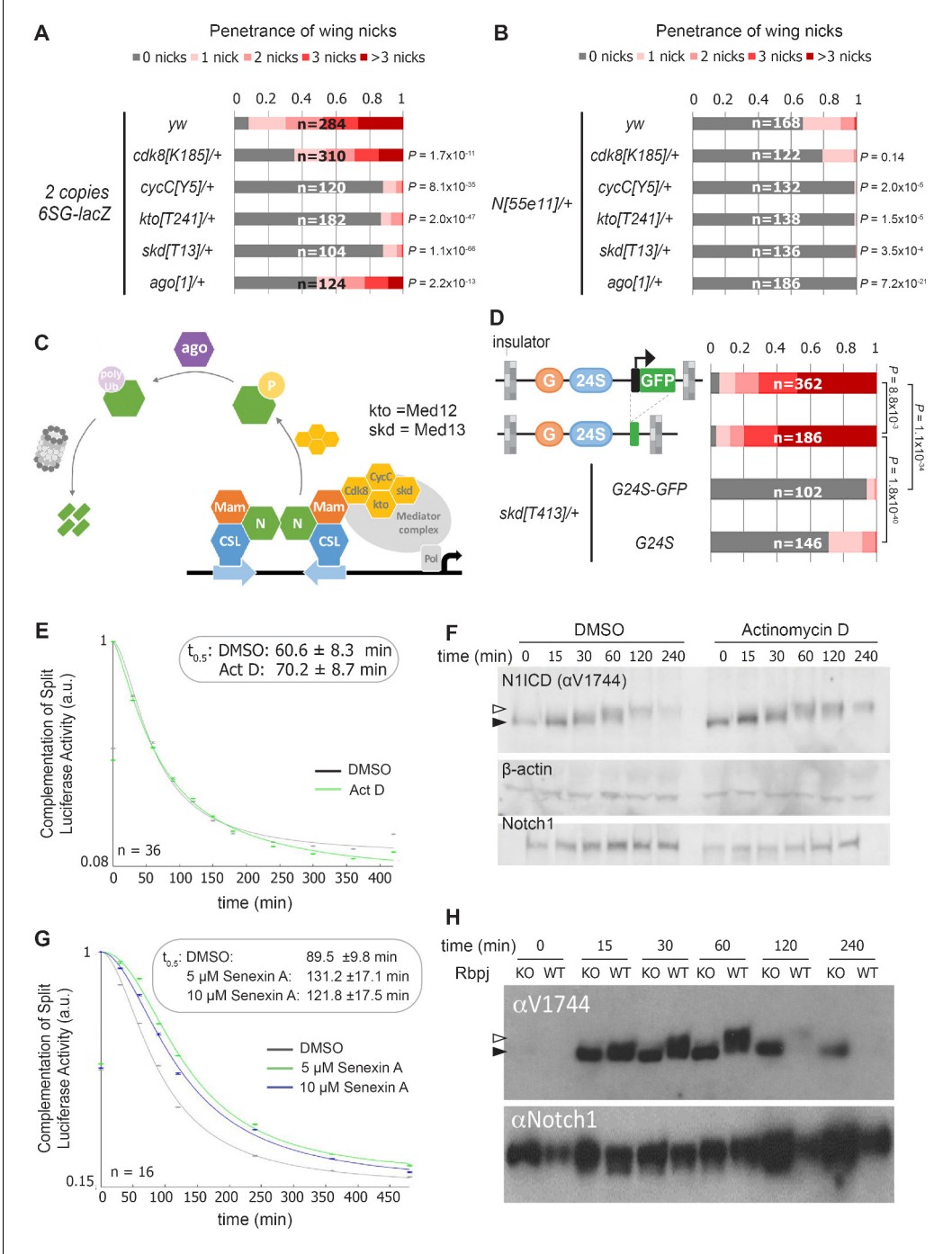

**Figure 3.** Reducing the activity of the Cdk8-Mediator suppresses the formation of wing notches. (**A–B**) Quantified wing notching in *6SG-lacZ* (**A**) or *N*<sup>55e11/+</sup> (**B**) in *wild-type* or *cdk8, cycC, kto, skd* or *ago* heterozygotes. Proportional odds model with Bonferroni adjustment tested for significance compared to wild-type. (**C**) Model of Cdk8-mediated NICD degradation. NCM complexes on SPSs recruit Cdk8, CycC, kto and skd. Cdk8 phosphorylates NICD to promote its degradation via Ago and the proteasome (gray cylinder). Cdk8 can also interact with the core Mediator (gray oval). (**D**) Schematics of promoter-containing and -lacking transgenes at left. Wing notching penetrance and severity at right. Proportional odds model was used to assess significance. (**E**) Rbpj-N1ICD split luciferase assay assessing N1ICD half-life in HEK293T cells treated with DMSO or Actinomycin D. 95% confidence interval noted. (**F**) Western blot of N1ICD, total Notch1 and β-actin after Notch activation in mK4 cells treated with DMSO or Actinomycin D. (**G**) Rbpj-N1ICD split luciferase assay assessing N1ICD half-life in HEK293T cells treated with DMSO, 5 μM Senexin A or 10 μM Senexin A. 95% confidence interval noted. (**H**)

*Figure 3 continued on next page*

*Figure 3 continued*

Western blot of N1ICD and full-length Notch1 after Notch activation in either *wild-type* (OT13) or *Rbpj*-deficient (OT11) cells. Open arrow denotes post-translationally modified NICD and closed arrow denotes un-modified NICD.

The online version of this article includes the following figure supplement(s) for figure 3:

**Figure supplement 1.** Decreased gene dose of the Cdk8-Mediator submodule suppresses the formation of wing nicks.

**Figure supplement 2.** Transcription inhibition doesn't impact NICD mobility.

---

flanked by insulator sequences (**Figure 3D**). We found that the wing notching penetrance and severity was similar with both transgenes, and the wing phenotype generated by both was significantly suppressed by removing an allele of *skd* (**Figure 3D**). These findings suggest that a transcriptionally active promoter is not required to induce wing nicks.

To test the generality of the idea that transcription activation could be uncoupled from NICD degradation, we blocked transcription using actinomycin-D and assessed NICD half-life using a split-luciferase assay in HEK293T mammalian cells (**Ilagan et al., 2011**). Importantly, we found that while actinomycin-D effectively inhibited Notch-induced transcription (**Figure 3—figure supplement 2A**), it neither altered N1ICD half-life in the split-luciferase assay (**Figure 3E**), nor altered N1ICD mobility in western blot analysis (**Figure 3F**). These data suggest that post-translational modification and degradation of N1ICD does not require active transcription. In contrast, inhibiting Cdk8 activity using Senexin-A or SEL120-34A, two structurally distinct and specific inhibitors of Cdk8 and Cdk19, a closely related vertebrate paralogue that is absent in *Drosophila* (**Porter et al., 2012**; **Rzymski et al., 2017**), significantly prolonged N1ICD half-life in the split-luciferase assay (**Figure 3G** and **Figure 3—figure supplement 2B**). Importantly, we found that N1ICD was stabilized in mammalian OT11 cells deficient for RBPJ (**Figure 3H**; **Kato et al., 1997**), in mK4 cells deficient for the three Mastermind-like (MAML) proteins, and in mK4 cells treated with SEL120-34A (**Figure 3—figure supplement 2C–E**). These data suggest that N1ICD degradation is coupled with NCM complex formation on DNA and CDK8-mediated modification. Moreover, the increased N1ICD mobility observed in the absence of RBPJ or MAML or in the presence of the SEL120-34A CDK8 inhibitor is consistent with a loss of post-translational modifications (**Figure 3H** and **Figure 3—figure supplement 2D,E**). Lastly, we found that treatment of protein extracts with Calf intestinal phosphatase (CIP) can abolish the mobility shift of NICD in wild type mK4 cells (**Valerius et al., 2002**) and to a lesser extent in SEL120-34A treated cells, whereas NICD in MAML knockout cells shows no change in mobility due to CIP treatment (**Figure 3—figure supplement 2F**). Altogether, these data indicate that Cdk8-mediated regulation of NICD degradation requires NCM complex formation on DNA in both mammalian cells and fly tissues but does not require active transcription.

## Quantitative analysis of enhancer binding site induced Notch turnover

Our data support a model whereby NCM binding to *SPS-GBE* sites promotes NICD phosphorylation and degradation, and thereby reduces NICD levels in the nucleus (**Figure 4A**). To obtain a quantitative understanding of how changes in SPS number affect Notch signal strength, we used mathematical modeling and quantitative expression analysis. The model includes a set of biochemical reactions that describe NICD dynamics in the nucleus (bottom, **Figure 4A**). We initially assume unphosphorylated, unbound NICD ($NICD_{up,ub}$) enters the nucleus at a constant production rate ($P_{NICD}$), where it forms NCM complexes that bind DNA. Bound, unphosphorylated NICD ($NICD_{up,b}$) can be phosphorylated by Cdk8 ($NICD_{p,b}$) at a rate $k_p$, assuming that NICD is not dephosphorylated by a phosphatase. Similar to the unphosphorylated state, phosphorylated NICD can cycle between $NICD_{p,b}$ and $NICD_{p,ub}$. Finally, it is assumed that the degradation rate of $NICD_{p,ub}$, denoted by $\Gamma_p$, is much faster than the degradation rate of $NICD_{up,ub}$, denoted by $\Gamma_{up}$.

To test these predictions, we measured Notch signal strength in wing margin cells in the presence and absence of *SPS-GBE* transgenes. To systematically vary SPS numbers, we created fly lines containing one, two or three (*3xGBE-6xSPS*) cassettes in front of a single *lacZ* gene ((*G6S*)*n-lacZ*). Analysis of flies carrying a single copy of the (*G6S*)*n-lacZ* transgenes revealed enhanced penetrance and severity of wing notching as the number of *G6S* cassettes increased, and all were significantly

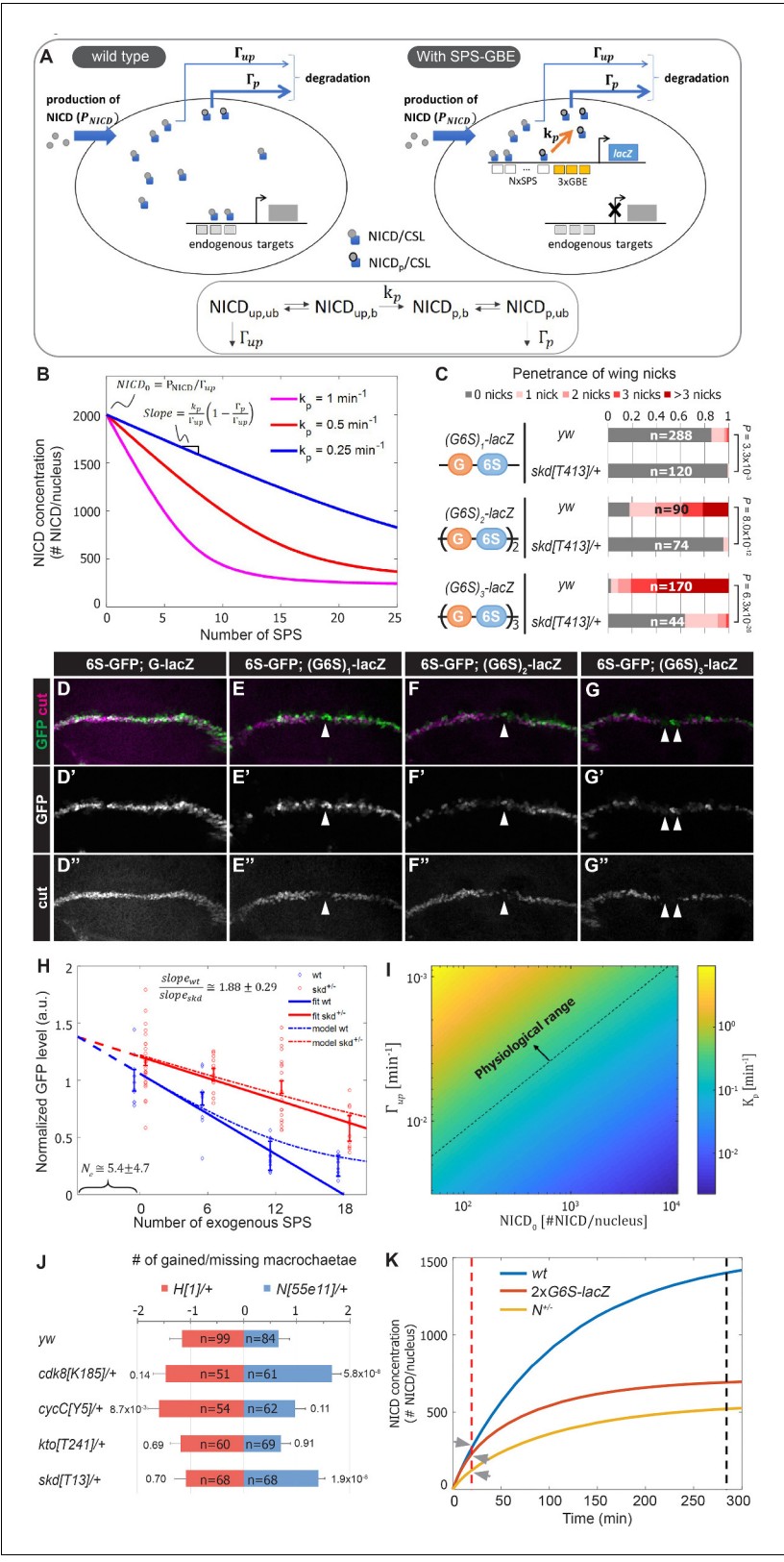

**Figure 4.** A mathematical model coupling NICD degradation to DNA binding predicts Notch activity and tissue sensitivity. (**A**) Schematic and equation describing *SPS-GBE* induced turnover of NICD. In both wild type (left) and nuclei with SPS-GBE sites inserted (right), NICD is produced and enters the nucleus at a constant rate ($P_{NICD}$). NCM complexes form on SPS, where NICD is phosphorylated by Cdk8 at a rate $k_p$. Phosphorylated NICD degrades faster ($\Gamma_p$) than unphosphorylated NICD ($\Gamma_{up}$). Subscripts $p, up, b, ub$ denote phosphorylated, unphosphorylated, bound, and unbound NICD. (**B**)

*Figure 4 continued on next page*

*Figure 4 continued*

Simulations of NICD levels as a function of SPS number. The three curves correspond to simulations with indicated values of $k_p$. NICD starts from a common level ($NICD_0$) and initially decreases linearly with SPS number, with a slope proportional to $k_p$. (C) Wing notching penetrance and severity in flies with indicated genotypes. Proportional odds model was used to assess significance. (D–G") Wing discs from flies containing *6xSPS-GFP* (*6S-GFP*) and either *GBE-lacZ* (*G-lacZ*) or *(G6S)$_{1,2 or 3}$-lacZ* stained with cut (magenta). (H) Quantified GFP levels in wing discs with increasing SPSs (0, 6, 12, 18 correspond to *(G6S)$_{1,2 or 3}$-lacZ*) in either wild type (blue) or *skd* heterozygotes (red). Each dot represents the average GFP level in margin cells from a single wing disc. Error bars show means and S.E.M for each disc. Solid lines represent linear fit to mean GFP values of the first three points of *wild-type* (blue) and the four points of *skd* heterozygotes (red). Ratio of slopes is indicated. Effective number of endogenous SPS, $N_e$, is estimated by extrapolating the y axis intersect of dashed lines. (I) Estimated phosphorylation rates by a single SPS, $k_p$. Values of $k_p$ (color-bar) were estimated for a range of values of $NICD_0$ and $\Gamma_{up}$. Dashed line represents lower limit of the physiological range of kinase activities. (J) Quantified number of gained/lost macrochaetae from indicated genotypes in $N^{55e11/+}$ (blue bars) or $H^{1/+}$ (red bars) background. Proportional odds model with Bonferroni adjustment. Data are mean ±95% confidence interval. (K) NICD level simulations as a function of time after Notch activation (at t = 0 min) in *wild-type* (blue), *N* heterozygotes (yellow) and *SPS-GBE* flies (red). In tissues with long duration Notch activation (black dash line), *N* heterozygotes and *SPS-GBE* sites similarly reduce NICD levels. In tissues with short duration Notch activation (red dash line), NICD levels are weakly affected by *SPS-GBE* compared to *N* heterozygotes (arrows).

The online version of this article includes the following figure supplement(s) for figure 4:

**Figure supplement 1.** *6S-GFP* reporter activity is sensitive to *Notch* gene dose.

**Figure supplement 2.** Dynamic model is robust for to variations in parameters.

suppressed by removing one *skd* allele (**Figure 4C**). Because direct measurement of nuclear NICD levels in vivo is very challenging (**Couturier et al., 2012**), we monitored NICD levels in wing margin cells indirectly via GFP expression from an independent *6xSPS-GFP* (*6S-GFP*) reporter that is highly sensitive to changes in *Notch* gene dose (**Figure 4—figure supplement 1**). We found that GFP levels decreased as a function of added *GBE-SPS* sites (**Figure 4D–H**). Simultaneous analysis of Cut, an endogenous *Notch* target required for maintaining wing margin fate (**Micchelli et al., 1997**; **Neumann and Cohen, 1996**), revealed a loss of wing margin fate in a subset of (G6S)$_2$-lacZ and (G6S)$_3$-*lacZ* cells (arrowheads in **Figure 4E–G**), consistent with the notched wing phenotype observed in these animals.

Analysis of the differential equations corresponding to these reactions generated several predictions. First, our model predicts steady state NICD levels will initially decrease linearly as the number of SPSs increases and then saturate for high numbers of SPSs (**Figure 4B**). Importantly, the linear regime of the slope describing NICD degradation is expected to be proportional to the Cdk8 phosphorylation rate, $k_p$. Accordingly, if there is no dosage compensation in CKM heterozygotes, the model's second prediction is that the slope of the wild-type curve should be twice that of the heterozygous mutant curve.

Analysis of *6S-GFP* expression revealed an approximately linear decrease in GFP as the number of *G6S* cassettes is increased (**Figure 4H**, blue markers). Moreover, removing an allele of *skd* significantly increased *6S-GFP* expression, resulting in a shallower slope relative to wild type flies with the same (G6S)-*lacZ* transgene (**Figure 4H**, red markers). The ratio between slopes as calculated by linear regression analysis of GFP levels in wild type and *skd* heterozygotes (solid lines in **Figure 4A**) was 1.88 ± 0.29, in agreement with the predicted 2-fold change in the absence of CKM dosage compensation. Interestingly, the two curves did not intersect at the y-axis, reflecting a cumulative reduction in NICD phosphorylation and degradation rates at endogenous sites; an interpretation supported by the observation that CKM heterozygotes ameliorate *Notch* heterozygote induced wing notching phenotypes (**Figure 3B**). We used this observation to estimate the magnitude of the cumulative genomic effect by extrapolating the crossing point of the two curves (dashed lines in **Figure 4H**). The lines crossed at negative $N_e = 5.4 \pm 4.7$ *SPS-GBE* sites. This value means that the cumulative effect on NICD stability of all sites in the genome ($N_e$) is equal to that of ~5 highly active synthetic *SPS-GBE* sites.

Next, we used the model to calculate Cdk8 phosphorylation rates, $k_p$, needed at *SPS-GBE* sites to lower NICD concentrations and induce wing notching phenotypes. In the linear regime, $k_p$ (in 1/min units) can be calculated from the measured slope, $Slope_{wt}$, for different values of $NICD_0$, $\Gamma_{up}$, and $\Gamma_p$ (see **Figure 4B** and Materials and methods for derivation). We used a plausible range of NICD concentrations, $NICD_0$, (between $10^2$-$10^4$ molecules per nucleus) and degradation rates $\Gamma_{up}$ (between $\frac{1}{30} min^{-1} - \frac{1}{1000} min^{-1}$, see Materials and methods for parameter estimation). This analysis

provided a range for likely Cdk8 catalytic activity on *SPS-GBE* sites between $10^{-1}$ to $10^{3}\ min^{-1}$, a large portion of which falls within the physiological regime of known kinase activities (above the dashed line in Figure 4I; *Davidi et al., 2016*; *Good et al., 2009*; *Kõivomägi et al., 2011*). In fact, the calculated $k_p$ values are at the low end of the physiological range (of the order of ~1/min), suggesting that even modest phosphorylation rates could produce the observed reduction in Notch signal strength, and ultimately, wing notching phenotypes induced by *SPS-GBE* sites.

## Altered NICD degradation sensitizes tissues requiring long duration signals

Molecularly, *Notch* haploinsufficiency is due to decreased NICD production, whereas our data support the model that the phenotype caused by *SPS-GBE* sites is due to enhanced NICD turnover. This difference in mechanism might explain why *SPS-GBE* sites fail to impact another known *N* dose sensitive tissue, the macrochaetae sensory bristles (*Figure 2K–L*). These data support the idea that wing margin cells are sensitive to changes in both NICD production and degradation, whereas macrochaetae formation is preferentially sensitive to changes in NICD production. To further test this hypothesis, we analyzed macrochaetae formation in compound heterozygotes for CKM genes and *Notch* or *Hairless*. In contrast to the observed suppression of wing notching (*Figure 3B*), we found that removing an allele of each CKM gene did not significantly suppress the *Notch* haploinsufficiency of extra macrochaetae (*Figure 4J*). In fact, macrochaetae numbers further increased in *N;cdk8* and *N;skd* compound heterozygotes, which is opposite of the predicted outcome if slowing NICD degradation significantly elevated NICD levels during macrochaetae formation (*Figure 4J*). As a second test to determine if changes in the CKM could alter macrochaetae, we analyzed *H* heterozygotes that generate too few macrochaetae due to increased Notch activity (*Morel et al., 2001*; *Maier et al., 1997*). In this genetic background, removing an allele of either *skd* or *cdk8* did not significantly alter macrochaetae formation (*Figure 4J*). However, we did find that removing an allele of *cycC* had a small, but significant impact on macrochaetae formation in *H;cycC* compound heterozygotes. Given that removing a *cycC* allele did not significantly impact macrochaetae formation in $N^{55e11/+}$ heterozygotes, it is possible that the decrease in macrochaetae numbers in *H;cycC* compound heterozygotes is due to changes in factors unrelated to the Notch pathway. Nevertheless, these data suggest that the Notch haploinsufficient phenotypes in macrochaetae lateral inhibition and wing margin formation are differently affected by decreasing CKM levels.

A potential explanation for the tissue-specific response to *SPS-GBE* sites and CKM heterozygotes could be the distinct temporal requirements for Notch activation in each tissue. Maintenance of wing margin identity is a continuous process at least 48 hours long (*de Celis et al., 1996*; *Shellenbarger and Mohler, 1978*), whereas macrochaetae formation requires Notch input over a short time period (< 30 min) (*Barad et al., 2010*). To explore the relationship between Notch signal duration and sensitivity to changes in NICD production/degradation rates, we modeled the dynamics of NICD accumulation as a function of time in wild-type, *N* heterozygotes, and flies homozygous for *G6S-lacZ* (*Figure 4K*). We assume that *N* heterozygotes lower NICD production ($P_{NICD}$) by one half without impacting NICD degradation, whereas *SPS-GBE* sites do not affect $P_{NICD}$ but increase NICD degradation as a function of SPS number. In a scenario where nuclear NICD reaches steady state (*Figure 4K*, black dash line), both the *SPS-GBE* loci and *N* heterozygotes significantly decrease NICD levels. In contrast, if Notch signals are only required for a short time period, changes in degradation rates do not significantly alter NICD levels relative to the impact of losing a *N* allele (*Figure 4K*, arrows). We note that this conclusion is robust over a broad range of potential NICD production and degradation rates (*Figure 4—figure supplement 2*). Moreover, the model is consistent with the results observed using genetic changes in CKM gene dose – altering NICD degradation selectively impacts long duration events (wing margin) and not short duration events (lateral inhibition during macrochaetae specification).

## Defining enhancer TF binding sites (TFBSs) that induce the notched wing phenotype

The synthetic Notch binding sites used in the GBE-SPS transgene were designed to minimize the inclusion of sequences bound by other known TFs (see Materials and methods). To test if an endogenous SPS sequence could induce wing phenotypes, we selected a previously characterized *E(spl)m8*

SPS sequence (*Furriols and Bray, 2001*; *Bailey and Posakony, 1995*; *Lecourtois and Schweisguth, 1995*). The *E(spl)m8* SPS is flanked by adjacent N-box sites bound by E(spl)/Hes factors (*Kramatschek and Campos-Ortega, 1994*), which thereby provide negative feedback downstream of Notch signaling (*Figure 5A*). Comparative analysis of flies carrying two copies of the synthetic *G6S(syn)-lacZ* versus the *G6Sm8-lacZ* revealed a highly similar penetrance and severity of wing notching phenotypes (*Figure 5B*). Thus, an endogenous SPS sequence from a known Notch-

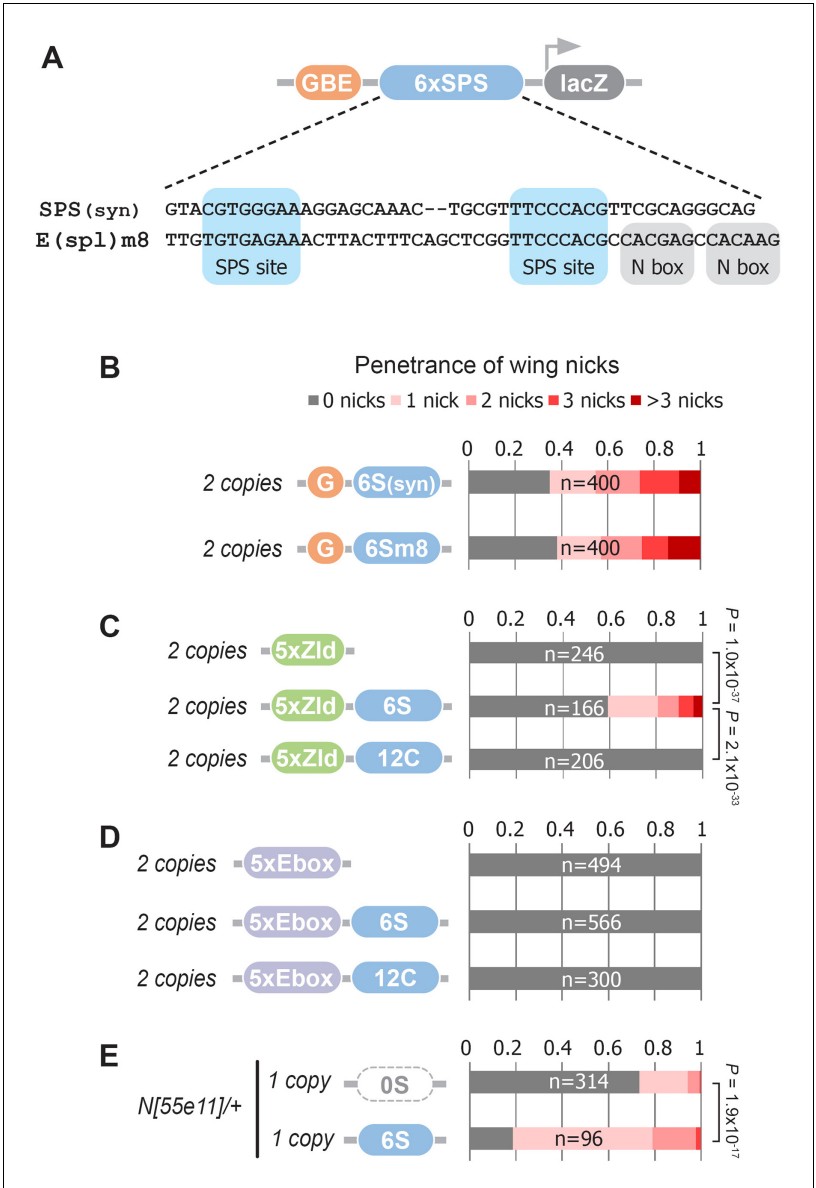

**Figure 5.** Defining enhancer TFBSs that induce the notched wing phenotype. (**A**) Graphical representation and sequence alignment of the synthetic SPS (SPS(syn)) and E(spl)m8 SPS constructs. SPS sites are highlighted in blue and Nbox sequences, which are only present in E(spl)m8, are shaded in gray. (**B**) Quantified wing notching in flies with two copies of either *G6Ssyn-lacZ* or *G6Sm8-lacZ* inserted in the same locus. (**C**) Quantified wing notching in flies with two copies of either *5xZelda(Zld)-lacZ*, *5xZld-6xSPS-lacZ*, or *5xZld-12xCSL-lacZ*. Two-sided Fisher's exact test assessed for significance between genotypes. (**D**) Quantified wing notching in flies with two copies of either *5xEbox-lacZ* sites alone, *5xEbox-6xSPS-lacZ*, or *5xEbox-12xCSL-lacZ*. (**E**) Quantified wing notching in *N* heterozygous flies containing a single copy of either *0xSPS-lacZ* or *6xSPS-lacZ* transgenes. Proportional odds model tested for penetrance/severity.

regulated enhancer can promote NICD degradation to induce a wing notching phenotype in the GBE-SPS transgene assay, even in the presence of potential negative feedback.

Next, we tested if additional TFs besides Grh can synergize with SPS sites to induce phenotypes by replacing the GBE sites with either Zelda (Zld) sites or an E-box sequence that binds basic-Helix-Loop-Helix (bHLH) TFs such as the *Drosophila* E-protein Daughterless (Da). Like Grh, Zld is a pioneer factor that opens chromatin (*McDaniel et al., 2019*), whereas the widely expressed Da E-protein has not been shown to have pioneering activity. Accordingly, we found that flies containing two copies of *5xZld-6xSPS-lacZ* induced notched wings, whereas flies with *5xZld-lacZ* or *5xZld-12xCSL-lacZ* failed to induce wing notching (*Figure 5C*). In contrast, 5xEbox sites failed to induce wing phenotypes when coupled to either SPS or CSL sites (*Figure 5D*). Taken together, these findings suggest that TFs with pioneering activity synergize with the SPS Notch dimer sites, but not the CSL Notch monomer sites, to promote Notch degradation.

The finding that pioneer TF sites are necessary to induce SPS-dependent wing phenotypes in wild type flies could be due to pioneer TFs either being strictly required to promote NICD turnover or pioneer TFs could accelerate this activity, perhaps by increasing genome accessibility. To distinguish between these possibilities, we tested if SPS enhancers lacking pioneer TF sites are capable of promoting Notch-dependent wing nicking in a sensitized genetic background. Importantly, we found that *Notch* heterozygous flies carrying a single copy of the *6S-lacZ* transgene had significantly enhanced wing notching compared to *Notch* heterozygous flies with an enhancer-less *lacZ* transgene (*0S-lacZ*) (*Figure 5E*). Taken together with the finding that *G6S-lacZ* and *5xZld-6S-lacZ* were sufficient to induce notched wings in wild type flies (*Figure 1K* and *Figure 5C*) whereas *6S-lacZ* failed to do so (*Figure 1—figure supplement 2*), these data suggest that enhancers with only SPS sites can promote NICD degradation, but to a lesser degree than those near a bound pioneer TF.

## Discussion

Our results show that simply increasing the number of clustered Notch dimer sites (SPS) linked to sites for a pioneer TF can cause a tissue-specific *N* haploinsufficiency phenotype via a Cdk8-dependent mechanism. These findings have important implications for both enhancer biology and the mechanisms regulating Notch signal strength in specific tissues. First, the proposed Cdk8-dependent mechanism links the rapid degradation of the Notch signal (NICD) with its binding to specific loci (SPSs) in a manner that can be uncoupled from transcription activation. This 'bind and discard' mechanism reveals an unexpected global link between accessible binding sites in the epigenome, such that the collective 'drain' loci can reduce Notch-dependent transcription at other loci in the same nucleus. Moreover, since the binding sites do not have to be coupled with transcription to induce a notched wing phenotype, our findings highlight the possibility that seemingly non-functional genomic binding events could impact TF metabolism in a Cdk8-dependent manner. Given that Cdk8 interacts with many genomic loci (*Pelish et al., 2015*) and that a previous phospho-proteomic study identified numerous transcriptional regulators are targets of CDK8/19 phosphorylation (*Poss et al., 2016*), such a mechanism may be quite general and apply to transcription regulators beyond Notch.

While our genetic and cell culture data, as well as previous phosphorylation studies, support a direct link between CKM activity and NICD degradation (*Fryer et al., 2004*), it should be noted that the CKM also phosphorylates other proteins that could contribute to the differences in wing versus macrochaetae phenotypes (*Poss et al., 2016*). Intriguingly, one of the high confidence CKM targets in mammalian cells was MAML1, indicating that CKM activity may directly regulate Notch output by phosphorylating multiple components of the Notch transcription complex. In addition, CCNC (CycC), MED12, and MED13 were all found to be high confidence CDK8/19 targets (*Poss et al., 2016*), suggesting that CKM activity may directly impact the turnover of key components of its own complex. It's thus not surprising that removing an allele of *cycC*, *kto* (*med12*) or *skd* (*med13*) had a much larger impact on the *Notch* haploinsufficient wing phenotype compared to changing the gene dose of *cdk8*. Moreover, these genetic data are consistent with prior studies in yeast showing that structural/regulatory components of macromolecular complexes, such as CycC/Med12/Med13, are enriched in haploinsufficiency genes, whereas enzymes are generally under-represented from the list of dose sensitive genes (*Kondrashov and Koonin, 2004*).

Second, our data support the idea that not all Notch binding sites are equally capable of marking NICD for degradation, and that enhancer architecture plays a key role in modulating NICD turnover. For instance, only Notch dimer but not Notch monomer sites are sufficient to generate phenotypes, and even SPS-containing enhancers differ in their ability to induce phenotypes based on the absence/presence of pioneer TF sites. Notably, we found that enhancers with either synthetic SPS sites designed to limit additional TF input or an endogenous *E(spl)m8* SPS with adjacent binding sites capable of providing negative feedback were sufficient to induce notched wing phenotypes when coupled to pioneer TF sites. Since Grh and Zld binding is sufficient to increase chromatin opening (*McDaniel et al., 2019*; *Jacobs et al., 2018*), these findings suggest enhancer accessibility alters the rate at which NICD is metabolized by Notch dimer sites. Intriguingly, ChIP-seq data for Grh (*Nevil et al., 2017*) and Zld (*Harrison et al., 2011*) reveals extensive binding to the *Enhancer of Split* (*E(spl)*) locus that contains numerous SPS-containing *Notch* regulated enhancers (*Cave et al., 2011*; *Cooper et al., 2000*; *Nellesen et al., 1999*). However, it is important to note that while an SPS-containing enhancer lacking pioneer TF sites failed to induce phenotypes in wild type flies, it did significantly increase wing notching in a sensitized genetic background. These findings suggest that SPS-containing enhancers promote NICD degradation at differing rates based on the presence of nearby TF sites. While the mechanistic basis for how SPS but not CSL sites promote NICD degradation is not known, these data highlight a potential mechanism by which enhancer architecture (i.e. Notch dimer vs monomer sites) and epigenetic 'context' (i.e. accessibility due to pioneer TF binding) can fine tune the global Notch response in different tissues.

The finding that introducing as few as 12 SPS sites into the genome can induce notched wing phenotypes raises the question of how many functional SPS sites exist in the endogenous genome. Recent studies found that about one third of direct *Notch* target genes (38 of 107 genes) in human T-ALL are regulated by SPS sites (*Severson et al., 2017*), and a mouse mK4 cell line has an estimated 2500 Notch dimer dependent binding sites (*Hass et al., 2015*). These findings suggest that many SPS sites are accessible across the mammalian genome. The estimated number of SPS sites within the *Drosophila* genome, which is an order of magnitude smaller than most mammalian genomes, remains to be determined. However, of the 154 Notch-responsive genes identified in a *Drosophila* wing disc-derived cell line, eight encode *E(spl)* genes that are clustered within a common 40 kb locus and many *E(spl)* genes contain one or more SPS sites (*Housden et al., 2013*). In comparison to the *E(spl)* locus, the 6 SPS sites within the *G6S-lacZ* transgene are found within ~300 bps, and thus it is possible that concentrating SPS sites might provide an avidity impact that increases the probability of a recruited NICD molecule being marked for degradation. In fact, we estimated that the cumulative effect of the *Drosophila* genome is equal to ~5 highly accessible, linked *SPS-GBE* sites. Future studies using endogenous SPS-containing enhancers will be needed to provide a better understanding of both the role of nearby binding sites for other TFs and how concentrating SPS sites in specific loci impacts the wing notching phenotype.

Third, we propose that the differential sensitivity of Notch-dependent tissues to changes in NICD degradation (i.e. *SPS-GBE* sites or CKM heterozygotes) or production rates (*N* heterozygotes) reflects the temporal requirement for Notch signal duration. An appealing aspect of this Notch signal duration model is that it predicts that any perturbation that alters NICD signal degradation will preferentially affect long-duration processes over short duration processes, whereas perturbations that impact NICD signal production will affect both long and short duration events. Moreover, the differential sensitivity of the Notch duration model to changes in production versus degradation rates may be generalizable to the study of other signaling pathways. However, it's worth noting that additional differences between the wing and macrochaetae besides signal duration may contribute to the magnitude of change in Notch signal strength in each tissue. As an example, *cis*-inhibition, which determines the fraction of functional Notch receptors on the cell membrane (*Lee et al., 2017*; *Pitsouli and Delidakis, 2005*; *Zeng et al., 1998*), could play a cell-specific role in modulating NICD production to a larger degree in one tissue over another. Taking this mechanism into consideration, the assumption that *Notch* heterozygotes reduce NICD production by 50 percent in each tissue may be over-simplified. Thus, further experiments using a system that is amenable to systematic changes in the length of Notch signal induction are needed to thoroughly test the signal duration model in multiple tissues.

Intuitively, the duration model suggests a mechanism underlying cell-specific context that may have implications for both developmental processes and tumorigenesis. For example, mutations in

the NICD PEST domain that decouple DNA binding and degradation are common in T-ALL (*Weng et al., 2004*), and CycC (*CCNC*) functions as a haploinsufficient tumor suppressor gene in T-ALL, at least in part, by stabilizing NICD (*Li et al., 2014*). These findings suggest that T-ALL is highly sensitive to alterations in NICD degradation. Indeed, T-ALL cells are 'addicted' to Notch and are thus dependent on a long duration signal (*Severson et al., 2017*). As ~30% of Notch target genes in T-ALL use SPS containing enhancers, our findings provide insight into how Notch PEST truncations and *CCNC* heterozygotes could each promote tumorigenesis by slowing CKM-mediated NICD turnover on SPS enhancers. Future studies focused on enhancers that recruit the CKM and other Notch-dependent cellular processes will help reveal how the temporal requirements for nuclear activities contributes to both normal development and disease states.

# Materials and methods

## Key resources table

| Reagent type (species) or resource | Designation | Source or reference | Identifiers | Additional information |
|---|---|---|---|---|
| Genetic reagent (*D. melanogaster*) | BAC{Notch} | Bloomington *Drosophila* Stock Center | BDSC:38665; RRID:BDSC_81271 | FlyBase symbol: PBac{N-GFP.FLAG} VK00033 |
| Genetic reagent (*D. melanogaster*) | N[55e11] | Bloomington *Drosophila* Stock Center | BDSC:28813; RRID:BDSC_28813 | |
| Genetic reagent (*D. melanogaster*) | H[1] | Bloomington *Drosophila* Stock Center | BDSC:515; RRID:BDSC_515 | |
| Genetic reagent (*D. melanogaster*) | kto[T241] | Bloomington *Drosophila* Stock Center | BDSC:63126; RRID:BDSC_63126 | |
| Genetic reagent (*D. melanogaster*) | kto[T631] | Bloomington *Drosophila* Stock Center | BDSC:63125; RRID:BDSC_63125 | |
| Genetic reagent (*D. melanogaster*) | skd[T13] | Bloomington *Drosophila* Stock Center | BDSC:63123; RRID:BDSC_63123 | |
| Genetic reagent (*D. melanogaster*) | skd[T413] | Bloomington *Drosophila* Stock Center | BDSC:63124; RRID:BDSC_63124 | |
| Genetic reagent (*D. melanogaster*) | cdk8[K185] | PMID:11171343 | | |
| Genetic reagent (*D. melanogaster*) | cycC[Y5] | PMID:11171343 | | |
| Genetic reagent (*D. melanogaster*) | ago[1] | PMID:11565033 | | |
| Genetic reagent (*D. melanogaster*) | ago[3] | PMID:11565033 | | |
| Cell line (*H. sapiens*) | HEK293T | PMID:23806616 | | Stable expression of CLuc-RBPjK and Notch1-NLuc |
| Cell line (*M. musculus*) | mK4 | PMID:11850199 | RRID:CVCL_9T80 | |
| Cell line (*M. musculus*) | OT-13 | PMID:9374409 | RRID:CVCL_T371 | wild-type embryonic fibroblast |
| Cell line (*M. musculus*) | OT-11 | PMID:9374409 | RRID:CVCL_T370 | RBPjK deficient embryonic fibroblast |
| Antibody | anti-NICD Cleaved Notch1 Val1744 D3B8 (Rabbit monoclonal) | CST | Cat # 4147; RRID:AB_2153348 | WB (1:1000) |

*Continued on next page*

*Continued*

| Reagent type (species) or resource | Designation | Source or reference | Identifiers | Additional information |
|---|---|---|---|---|
| Antibody | anti-Notch1 Clone D1E11 (Rabbit monoclonal) | CST | Cat# 3608; RRID:AB_2153354 | WB (1:1000) |
| Antibody | anti-cut (Mouse monoclonal) | DSHB | Cat# 2B10 | IF (1:50) |
| Antibody | anti-β-Actin Clone AC-15 (Mouse monoclonal) | Sigma-Aldrich | Cat# A5441; RRID:AB_476744 | WB (1:4000) |
| Antibody | anti- MAML1 D3E9 (Rabbit monoclonal) | CST | Cat# 11959; RRID:AB_2797778 | WB (1:1000) |
| Antibody | anti-MAML2 (Rabbit polyclonal) | CST | Cat# 4618; RRID:AB_2139273 | WB (1:1000) |
| Antibody | anti-MAML3 (Rabbit polyclonal) | Bethyl | Cat# A300-684A; RRID:AB_2266032 | WB (1:1000) |
| Antibody | anti-RUNX1 D4A6 (Rabbit monoclonal) | CST | Cat# 8529; RRID:AB_10950225 | WB (1:1000) |
| Antibody | ECL anti-rabbit-HRP | GE Healthcare | Cat# NA934; RRID:AB_772206 | WB (1:5000) |
| Antibody | ECL anti-mouse-HRP | GE Healthcare | Cat# NA931; RRID:AB_772210 | WB (1:5000) |
| Sequence-based reagent | MAML1 exon1 PX458 F | This paper | guide RNA | CACCGCCGAAGTGGCAGCCGGCGCC |
| Sequence-based reagent | MAML1 exon1 PX458 R | This paper | guide RNA | AAACGGCGCCGGCTGCCACTTCGGC |
| Sequence-based reagent | MAML1 exon1 PX459 F | This paper | guide RNA | CACCGCGCCGGAAGAGGCGTTTTC |
| Sequence-based reagent | MAML1 exon1 PX459 R | This paper | guide RNA | AAACGAAAACGCCTCTTCCGGCGC |
| Sequence-based reagent | MAML2 exon1 PX458 F | This paper | guide RNA | CACCGGGGGCCTCCCAGTAAATAA |
| Sequence-based reagent | MAML2 exon1 PX458 R | This paper | guide RNA | AAACTTATTTACTGGGAGGCCCCC |
| Sequence-based reagent | MAML2 exon1 PX459 F | This paper | guide RNA | CACCGACTCCCACCAGTGATTAGTT |
| Sequence-based reagent | MAML2 exon1 PX459 R | This paper | guide RNA | AAACAACTAATCACTGGTGGGAGTC |
| Sequence-based reagent | MAML3 exon1 PX458 F | This paper | guide RNA | CACCGCTCCCGGGGCACACTATTT |
| Sequence-based reagent | MAML3 exon1 PX458 R | This paper | guide RNA | AAACAAATAGTGTGCCCCGGGAGC |
| Sequence-based reagent | MAML3 exon1 PX459 F | This paper | guide RNA | CACCGCTCACTGGGGTGCGCGTTG |
| Sequence-based reagent | MAML3 exon1 PX459 R | This paper | guide RNA | AAACCAACGCGCACCCCAGTGAGC |
| Chemical compound, drug | Actinomycin D | Sigma-Aldrich | Cat# A1410 | |
| Chemical compound, drug | Senexin A | R and D | Cat# 4875 | |
| Chemical compound, drug | SEL120-34A | Medchemexpress | Cat# HY-111388A | |
| Chemical compound, drug | CIP | NEB | Cat# M0525S | |

*Continued on next page*

*Continued*

| Reagent type (species) or resource | Designation | Source or reference | Identifiers | Additional information |
|---|---|---|---|---|
| Chemical compound, drug | D-Luciferin | Goldbio | Cat# LUCK-100 | |
| Chemical compound, drug | Trypsin-EDTA | Gibco | Cat# 25300–120 | |
| Chemical compound, drug | SuperSignal Femto West Chemoluminescent Substrate | Thermo-Fisher Scientific | Cat# PI34095 | |
| Software, algorithm | MATLAB | Mathworks | RRID:SCR_001622 | Codes available at https://github.com/ OhadGolan/NICD-concentration-in-the -nucleus-as-by-binding -site-coupled-NICD-degradation |

## Reporter design, molecular cloning, and transgenic fly generation

All synthetic Notch enhancer sequences contain high affinity Su(H) binding sites (CGTGGGAA). Notch monomer sites (CSL) were placed 17bps apart in a head-to-tail manner to permit independent binding of NCM complexes. Notch dimer sites (SPS) were spaced 15bps apart in a head-to-head orientation to enable cooperative dimerization between adjacent NCM complexes. Intervening sequences were designed to exclude known binding sites for other *Drosophila* TFs using the *cisBP* website (*Weirauch et al., 2014*). The 6xSPSmut sequence is identical to 6xSPS except for two nucleotide changes in each site in positions previously shown to disrupt CSL binding (CG**A**GG**C**AA) (*Tun et al., 1994*). The 2xSPS, 6xSPS, 6xSPSmut, 12xCSL, 6xSPSm8, 5xZelda and 5xEbox sequences were synthesized by GenScript as either complementary oligonucleotides (2xSPS) or double stranded DNA (6xSPS, 6xSPSmut, 12xCSL, 6xSPSm8, 5xZelda, and 5xEbox; complete sequences listed below). Cloning was facilitated by including flanking EcoR1 and HindIII/BglII sequences. Annealed oligonucleotides or double stranded DNA fragments were cloned into either *placZ-attB* or *3xGBE-placZ-attB* (*Uhl et al., 2016*) and sequence confirmed. To concatenate 6xSPS into larger arrays, a shuttle vector was used to generate a BamH1-6xSPS-BglII-Not1 fragment for reiterative cloning into vectors digested with BglII/NotI (BglII/NotI permits cloning BamH1/NotI fragments, which can be repeated as desired). To create the *3xGBE-24xSPS-lacZ* vector, a *24xSPS* fragment (generated in the shuttle vector) was cloned into the *3xGBE-lacZ* vector. To make the promoter containing *3xGBE-24xSPS-GFP* vector, we inserted *3xGBE-24xSPS* into *pHStinger-attB* (*Barolo et al., 2000*). To generate a promoterless *3xGBE-24xSPS* construct, the promoter and GFP encoding sequences were removed from *3xGBE-24xSPS-GFP* by KpnI/SpeI digest, blunted with T4 DNA polymerase, and ligated. To generate the (*3xGBE-6xSPS*)n-*lacZ* (n = 2,3) constructs, the *3xGBE-6xSPS* fragment was iteratively cloned into the *3xGBE-6xSPS-lacZ* plasmid. The *6xSPS-GFP* reporter used to measure Notch transcription responses was generated by cloning 6xSPS into the *pHStinger-attB* vector. All transgenic fly lines were generated by phiC31 recombinase integration into 22A, 51C or 86Fb loci of the *Drosophila* genome (Rainbow Transgenic Flies, Inc) with the transgene insertion location for each experiment listed in *Supplementary file 1*.

## Synthetic enhancer DNA and probe design

Enhancer sequences used in the transgenic reporter vectors and the DNA probes used in electromobility shift assays (EMSAs) are listed in FASTA format. Sequences are annotated as following: Restriction enzyme sites (RE) and/or RE overhangs (italics), Su(H) binding sites (blue), Zelda and Ebox binding sites (purple), Nbox (red) and point mutations (bold and underlined).

>6xSPS
*GAATTC*AGCTACGTGGGAAAGGAGCAAACTGCGTTTCCCACGTTCGCAGGGCAGCTA
CGTGGGAAAGGAGCAAACTGCGTTTCCCACGTTCGCAGGGCAGCTACGTGGGAAAG-
GAGCAAACTGCGTTTCCCACGTTCGCAGGGCAGCTACGTGGGAAAGGAGCAAACTGCG

TTTCCCACGTTCGCAGGGCAGCTACGTGGGGAAAGGAGCAAACTGCGTTTCCCACGTTCG-
CAGGGCAGCTACGTGGGAAAGGAGCAAACTGCGTTTCCCACGTTCGCAGGGC*AGATCT*

>6xSPS-mut
*GAATTC*AGCTACG**A**GG**C**AAAGGAGCAAACTGCGTTT**G**CC**T**CGTTCGCAGGGCAGCTA
CG**A**GG**C**AAAGGAGCAAACTGCGTTT**G**CC**T**CGTTCGCAGGGCAGCTACG**A**GG**C**AAAG-
GAGCAAACTGCGTTT**G**CC**T**CGTTCGCAGGGCAGCTACG**A**GG**C**AAAGGAGCAAACTGCG
TTT**G**CC**T**CGTTCGCAGGGCAGCTACG**A**GG**C**AAAGGAGCAAACTGCG
TTT**G**CC**T**CGTTCGCAGGGCAGCTACG**A**GG**C**AAAGGAGCAAACTGCG
TTT**G**CC**T**CGTTCGCAGGGC*AGATCT*

>12xCSL
*GAATTC*GCCCTGCGAACGTGGGGAAACCTAGGCTAGAGGCACCGTGGGAAACTGCC
TGCCCTGCGAACGTGGGAAACCTAGGCTAGAGGCACCGTGGGAAACTGCCTGCCC
TGCGAACGTGGGAAACCTAGGCTAGAGGCACCGTGGGAAACTGCCTGCCCTGCGAA
CGTGGGAAACCTAGGCTAGAGGCACCGTGGGAAACTGCCTGCCCTGCGAA
CGTGGGAAACCTAGGCTAGAGGCACCGTGGGAAACTGCCTGCCCTGCGAA
CGTGGGAAACCTAGGCTAGAGGCACCGTGGGAAACTGCCT*AGATCT*

>2xSPS_Cloning_oligonucletide#1
*aattc*AGCTACGTGGGAAAGGAGCAAACTGCGTTTCCCACGTTCGCAGGGCAGCTA
CGTGGGAAAGGAGCAAACTGCGTTTCCCACGTTCGCAGGGC*a*

>2xSPS_Cloning_oligonucleotide#2
*gatct*GCCCTGCGAACGTGGGAAACGCAGTTTGCTCCTTTCCCACGTAGCTGCCCTGCGAA
CGTGGGAAACGCAGTTTGCTCCTTTCCCACGTAGCT*g*

>5xZelda
*AAGCTT*TGCAGGTAGACGCAGTTTGCTCCTGCAGGTAGTAGCTGCCCTGCGATGCAGG-
TAGACGCAGTTTGCTCCTGCAGGTAGTAGCTGCCCTGCGATGCAGGTAG*GAATTC*

>5xEbox
*AAGCTT*GCCAGGTGTACGCAGTTTGCTCCTGCCAGGTGTTAGCTGCCTGCGAAG
CCAGGTGTACGCAGTTTGCTCCTGCCAGGTGTTAGCTGCCTGCGAAG
CCAGGTGT*GAATTC*

>6xSPSm8
*GAATTC*AGCTTGTGTGAGAAACTTACTTTCAGCTCGGTTCCCACGCCACGAGCCA-
CAAGTTGTGTGAGAAACTTACTTTCAGCTCGGTTCCCACGCCACGAGCCACAAGTTGTGT-
GAGAAACTTACTTTCAGCTCGGTTCCCACGCCACGAGCCACAAGTTGTGTGAGAAAC
TTACTTTCAGCTCGGTTCCCACGCCACGAGCCACAAGTTGTGTGAGAAACTTACTTTCAGC
TCGGTTCCCACGCCACGAGCCACAAGTTGTGTGAGAAACTTACTTTCAGCTCGG
TTCCCACGCCACGAGCCACAAG*AGATCT*

>1xSPS_EMSA_oligonucleotide
GCTACGTGGGAAAGGAGCAAACTGCGTTTCCCACGTTCGTAGTGCGGGCGTGGCT

>2xCSL_EMSA_oligonucleotide
CGAACGTGGGAAACCTAGGCTAGAGGCACCGTGGGAAACTAGTGCGGGCGTGGCT

>EMSA_5'IRDye-700/800_complementary_oligonucleotide
AGCCACGCCCGCACT

## Fly husbandry

The following alleles were obtained from the Bloomington *Drosophila* Stock Center: PBac{N-GFP.
FLAG}VK00033 (stock #38665), N[55e11] (#28813), H[1] (#515), kto[T241] (#63126), kto[T631]
(#63125), skd[T13] (#63123) and skd[T413] (#63124). cdk8[K185] and cycC[Y5] alleles were gifts from
Professor *Treisman (2001)*. ago[1] and ago[3] alleles were previously described (*Moberg et al.,
2001*). Flies were maintained under standard conditions with all genetic crosses, phenotyping and
gene expression assays performed at 25℃. The detailed genetic crosses needed to generate the
progeny in each Figure are listed in *Supplementary file 1*.

## Genetic assays

To analyze the wing notching and macrochaetae phenotypes, flies of the appropriate genotypes were mated in cornmeal-containing vials and transferred to fresh food every day. During our studies, we observed that changes in food quality and overcrowding could change the severity/penetrance of wing phenotypes, introducing variation. Hence, all experiments quantifying wing phenotypes within each Figure panel contained control flies that were grown on the same batch of food and with a similar animal density. Offspring of the listed genotypes were selected and the number of nicks on each wing was recorded and/or the number of dorsocentral and scutellar macrochaetae was counted. A Fisher's exact test was used to determine significance between samples when penetrance was being assessed (i.e. no phenotype versus a phenotype), whereas a proportional odds model was used to determine significance when the analysis included phenotype severity.

For L5 wing vein length, fly wings of the proper genotypes were dissected, mounted on glass slides and imaged using a Nikon NiE upright widefield microscope. The total length of the presumptive L5 vein and the vein-missing gap were measured using Imaris software. Student's t-test was used to determine significance.

## GFP reporter assays in larval imaginal wing discs

To systematically assess Notch transcription responses in larval wing imaginal discs, animals homozygous for *6xSPS-GFP$_{22A}$* and either *3xGBE-lacZ*, *3xGBE-6xSPS-lacZ*, *(3xGBE-6xSPS)$_2$-lacZ*, or *(3xGBE-6xSPS)$_3$-lacZ* were mated to either *yw* (wild type) or *skd[T413]/TM6B* males. Imaginal discs from male non-TM6B wandering 3$^{rd}$ instar larvae (*skd[T413]* heterozygotes) were dissected and fixed in 4% formaldehyde for 15 min. Samples were subsequently washed 4 times with PBX (0.3% Triton X-100 in PBS) and incubated with an antibody that recognizes the Cut antigen (mouse 1:50, DSHB) followed by a fluorescent-conjugated secondary antibody (Alexa Fluor, Molecular Probes). For quantitative purposes, at least eight imaginal discs were analyzed for each genetic condition tested and the entire wild type imaginal disc series was harvested, fixed and imaged at the same time using a Nikon A1R inverted confocal microscope (40x objective). For the *skd* heterozygote series, a set of wild type imaginal discs with *3xGBE-lacZ* was performed simultaneously to normalize the responses between series. All imaging was performed with constant settings for GFP levels, and GFP pixel intensity in wing margin cells was determined from Z-stack images using Imaris software. Two-sided Student's t-test was used to determine significance between samples.

## Protein purification and electrophoretic mobility shift assay (EMSA)

For recombinant protein purification, constructs that correspond to mouse RBPJ (aa 53–474), mouse N1ICD (aa 1744–2113), human MAML1 (aa 1–280), fly Su(H) (aa 98–523), fly NICD (aa 1763–2412) and fly Mastermind (aa 87–307) were expressed and purified from bacteria using a combination of affinity (Ni-NTA or Glutathione), ion exchange, and size exclusion chromatography as previously described (*Friedmann et al., 2008*). Purified proteins were confirmed by SDS-PAGE with Coomassie blue staining and concentrations were measured by absorbance at UV280 with calculated extinction coefficients. EMSAs were performed as previously described using native polyacrylamide gel electrophoresis (*Uhl et al., 2016*; *Uhl et al., 2010*). Proteins concentrations for each gel are listed in figure legends. Acrylamide gels were imaged using the LICOR Odyssey CLx scanner.

## Split luciferase assay and half-life estimations

Stability of mammalian N1ICD was analyzed using the previously described split-luc HEK293T cells (*Liu et al., 2013*; *Ilagan and Kopan, 2014*; *Ilagan et al., 2011*). These cells, which were engineered, generated and continuously maintained by the Kopan lab, express unique fusion proteins (CLuc-RBPjK and NOTCH1-NLuc) that provide a luciferase complementation assay. The cell line was authenticated by the inducible activation of Notch leading to the successful recapitulation of published data using the Luciferase complementation assay. In this study, these HEK293T cells were cultured for 8 hr with 50 nM Actinomycin D to block transcription or 4 hr in the presence of the inhibitor Senexin A to block CDK8/CDK19-mediated phosphorylation. Alternatively, cells were incubated 1 hr with 1 µM SEL120-34A (Medchemexpress) to block CDK8/19 mediated phosphorylation. Each inhibitor was present throughout the entire time-course of each respective experiment. Cells were activated for 10 min with 0.05% Trypsin-EDTA or with Trypsin only as a negative control and

transferred to Poly-D-Lysine coated black 96-well plates with 40,000 cells/well. Cells were cultured at 37°C for 1 hr to let cells attach. Medium was changed to Opti-MEM (Gibco) with 50 nM ActD, 1 μM SEL120-34A or the indicated concentration of Senexin A, and 150 μg/ml of D-Luciferin (Goldbio) substrate was added fresh before each measurement. The first measurement (t = 0) was carried out 1 hr after activation. Luciferase signals were measured using the IVIS Lumina LT system and were normalized to Trypsin treated controls and each well separately to the signal counts at t = 0.

To confirm the activity of Actinomycin D, mK4 cells with a stably integrated a 6xSPS-NanoLuc reporter or a 0xSPS-NanoLuc construct as control were used. Cells were cultured for 8 hr with 50 nM Actinomycin D or 0.1% DMSO and activated for 10 min with 0.05% Trypsin-EDTA (Gibco). Nano-Luc activity was measured after 3 hr using the Nano-Glo Luciferase Assay (Promega) and imaged with the IVIS Lumina LT system.

Since the decay curves in Figure 3 did not exhibit simple exponential decay, we calculated the half-life values by fitting the luciferase activity values to a decreasing Hill function, $\frac{t_{0.5}^n + bg \cdot t^n}{t_{0.5}^n + t^n}$, where t is time, $t_{0.5}$ is the half life, $bg$ is the background level, and $n$ is the Hill coefficient. The fitting was performed using the least mean square fitting algorithm in MATLAB. The half-life values are presented with 95% confidence intervals.

## Western blot analysis

Mouse Kidney 4 (mK4) cells, which were developed by and a kind gift from Steve Potter's lab (*Valerius et al., 2002*), were routinely authenticated by transcriptomic (RNAseq) and functional genomic (ATAC-seq) studies. For Western blot analysis of mammalian N1ICD, mK4 cells were cultured 8 hr with 50 nM Actinomycin D (Sigma-Aldrich, A1410), 1 hr with 2 μM SEL120-34A or 0.1% DMSO in medium before activation of NOTCH with 0.05% Trypsin-EDTA (Gibco) for 10 min. Cells cultured in medium containing DMSO, Actinomycin D or SEL120-34A and harvested at different timepoints (t = 0 was taken 15 min after activation), lysed in RIPA for 30 min and sonicated. For the phosphatase treatment studies, lysates were mixed with the same amount of CIP-buffer (100 mM NaCl, 50 mM Tris-HCL pH 7.9, 10 mM MgCl$_2$, 1 mM DTT) and incubated with Calf intestinal phosphatase (CIP, NEB; 20,000U/1 × 10$^6$ cells) for 60 min at 37°C. Equal amounts were loaded on 6% Acrylamide gels for SDS-PAGE and blotted on Nitrocellulose membranes (GE healthcare). Membranes were blocked in 5% dry milk powder in PBS with 0.1% Tween and incubated with anti-N1ICD (Val1744, CST, 1:1000), anti-Notch1 (D1E11, CST, 1:1000) and anti-β-actin (Sigma-Aldrich, 1:4000) overnight at 4°C. After incubation with HRP-conjugated secondary antibodies (GE Healthcare, 1:5000), signals were detected using SuperSignal Femto West Chemoluminescent Substrate (Thermo Fisher Scientific) and imaged with the BioRad ChemiDoc system.

Mouse embryonic fibroblast cells deficient for RBP-JK (OT-11) (*Kato et al., 1997*) or wild-type control (OT-13) cells were collected at various times after trypsin/EDTA treatment from confluent wells of a 12 well plate. The OT-11 cells were authenticated by showing a lack of target gene responsiveness to Notch stimulation and via Western blot analysis showing a loss of RBPJ protein. The cells were washed with PBS, lysed in 100 μL of RIPA + protease inhibitors and 100 μL of 2X sample buffer, and DNA was sheared with a needle. Equivalent amounts of lysate were run on an SDS polyacrylamide gel, transferred to nitrocellulose, blotted for active (Val1744, CST, 1:1000) or total Notch1 protein (D1E11, CST, 1:1000), and developed as described above.

We generated mK4 cells deficient for all mastermind-like proteins through CRISPR Cas9 mediated deletion of exon 1 of Mastermind-like 1, 2, and 3. Briefly, parental mK4 cells were simultaneously transfected with PX458 and PX459 containing guide RNAs flanking exon1 for each of the Mastermind-like genes and subjected to selection with puromycin for 2 days. Surviving clones were picked and authenticated the following week using cloning disks and screened by PCR analysis and Western blot to identify clones that lacked expression of Mastermind-like 1, 2, and 3. The sequences of guide RNAs and genotyping PCR primers are in Key Resources Table. All cell lines used throughout these studies were tested negative for mycoplasma contamination and no cell lines were used from the list of commonly misidentified cell lines maintained by the International Cell Line Authentication Committee.

## Mathematical model

The mathematical model describes the concentrations of the possible states of NICD. These include unphosphorylated unbound NICD, $NICD_{up,ub}$, unphosphorylated bound, $NICD_{up,b}$, phosphorylated bound, $NICD_{p,b}$, and phosphorylated unbound, $NICD_{p,ub}$. The dynamic equations described by the set of biochemical reactions presented in **Figure 4a** are:

$$\frac{d}{dt}NICD_{up,ub} = P_{NICD} - NICD_{up,ub}\Gamma_{up} - NICD_{up,ub}N_{ub}k_\alpha^+ + NICD_{up,b}k_\alpha^- \tag{1}$$

$$\frac{d}{dt}NICD_{up,b} = NICD_{up,ub}N_{ub}k_\alpha^+ - NICD_{up,b}k_\alpha^- - NICD_{up,b}k_p \tag{2}$$

$$\frac{d}{dt}NICD_{p,b} = NICD_{up,b}k_p - NICD_{p,b}k_\alpha^- + NICD_{p,ub}N_{ub}k_\alpha^+ \tag{3}$$

$$\frac{d}{dt}NICD_{p,ub} = NICD_{p,b}k_\alpha^- - NICD_{p,ub}\Gamma_p - NICD_{p,ub}N_{ub}k_\alpha^+ \tag{4}$$

Here, $P_{NICD}$ is the rate NICD enters the nucleus (production rate), $\Gamma_p$ and $\Gamma_{up}$ are the degradation rates of phosphorylated/unphosphorylated NICD, respectively, $k_\alpha^+, k_\alpha^-$ are the association and dissociation rates of the NCM complex to an SPS site, $k_p$ is the Cdk8 phosphorylation rate of bound unphosphorylated NICD, and $N_{ub}$ is the number of unbound SPS sites.

The total nuclear NICD concentration, $NICD_{tot}$, is the sum of the phosphorylated, $NICD_p$, and unphosphorylated fractions, $NICD_{up}$:

$$NICD_{tot} = NICD_p + NICD_{up} \tag{5}$$

The total phosphorylated and unphosphorylated NICD are the sum of the bound (index b) and unbound (index ub) fractions:

$$NICD_p = NICD_{p,b} + NICD_{p,ub} \tag{6}$$

$$NICD_{up} = NICD_{up,b} + NICD_{up,ub} \tag{7}$$

Combining **Equations 1,2, 3, 4, 5, 6, 7** and assuming the system is in steady state gives:

$$0 = NICD_0 - NICD_{up} - NICD_{up,b}\left(k_p' - 1\right) \tag{8}$$

$$0 = NICD_{up,b}\,k_p' - NICD_p\Gamma_p' + NICD_{p,b}\Gamma_p' \tag{9}$$

where we define the dimensionless parameters: $NICD_0 \equiv \frac{P_{NICD}}{\Gamma_{up}}$, $\Gamma_p' \equiv \frac{\Gamma_p}{\Gamma_{up}}$, $k_p' = \frac{k_p}{\Gamma_{up}}$.

$NICD_{up,b}$ and $NICD_{p,b}$ are calculated using a physical model based on equilibrium statistical mechanics (**Brewster et al., 2014**). The conceptual basis of such models is that the occupancy of binding sites can be deduced by examining the equilibrium probabilities of binding and unbinding of TFs to TFBSs. In such models, each state of the system is denoted with a statistical weight ($S_i$). In equilibrium, the statistical weights can be represented as the ratio of the concentration of each binding species $[X]$, to the dissociation rate $k_d^{[X]}$ associated with that interaction so that $S_i = [X]/k_d^{[x]}$ (**White et al., 2012**). The partition function is defined as the summation of all possible statistical weights of the system:

$$Z = \sum S_i \tag{10}$$

The partition function is the normalization factor by which the probabilities of the different states of the system are calculated so that the probability for state j is:

$$prob(state\,j) = S_j/Z \tag{11}$$

For our model, the statistical weights of the states of bound unphosphorylated and phosphorylated NICD are:

$$\alpha_{up} = \frac{NICD_{up}}{k_\alpha} \tag{12}$$

$$\alpha_p = \frac{NICD_p}{k_\alpha} \tag{13}$$

where $k_\alpha = \frac{k_\alpha^-}{k_\alpha^+}$ is the dissociation constant of NCM to an SPS site. Thus, the partition function of one binding site is:

$$Z = 1 + \alpha_{up} + \alpha_p \tag{14}$$

The total number of SPS sites, $N$, is comprised of endogenous SPS sites ($N_e$) and synthetic SPS sites ($N_s$):

$$N = N_e + N_s \tag{15}$$

Note, $N_e$ is an effective number for the cumulative impact of all endogenous sites (i.e. the many weak Notch binding sites within the genome) relative to the effect of the strong SPS sites in the *GBE-SPS* transgenes.

Combining *Equations 9, 10, 11, 12, 13, 14, 15* results in the following two equations:

$$0 = NICD_0 - NICD_{up} - (N_e + N_s)\frac{NICD_{up}}{k_\alpha + NICD_p + NICD_{up}}\left(k_p' - 1\right) \tag{16}$$

$$0 = (N_e + N_s)\frac{NICD_{up}}{k_\alpha + NICD_p + NICD_{up}}k_p' - NICD_p\Gamma_p' + (N_e + N_s)\frac{NICD_p}{k_\alpha + NICD_p + NICD_{up}}\Gamma_p' \tag{17}$$

This statistical mechanics approach is based on three main assumptions: (1) The binding dynamics (on and off rates) are much faster than the dynamics determining the level of NICD in the nucleus. This is clearly valid as the DNA binding time scales are of the order of seconds (*Gomez-Lamarca et al., 2018*) and degradation time scales are of the order of minutes to hours. (2) The number of NICD molecules in the nucleus is larger than the number of SPS sites. Since we typically look at a range of NICD concentrations of $10^2 - 10^4$ per nucleus, and a maximum number of SPS sites of 36, this assumption is also justified. (3) For simplicity, we assume that binding to different SPS sites are independent.

We also consider the situation where the nuclear NICD concentration is much higher than the dissociation rate: $[NICD_p] + [NICD_{up}] \gg k_\alpha$, namely, that we are in a strong binding regime. Under this assumption, the results are largely independent of the values of $k_\alpha$.

We use *Equation (16) and (17)* to solve for $NICD_p$, $NICD_{up}$, and $NICD_{tot}$ and obtain their steady state levels for each set of parameters. These steady state solutions were used to plot Figures 4b and 4h (model curves).

## Analysis of the linear regime

Since phosphorylation of NICD by Cdk8 occurs only for bound NICD, it can be assumed that for a low number of SPS sites $NICD_p, k_\alpha, (N_e + N_s) \ll NICD_{up}$. In this regime, *Equations 16 and 17* are approximated by:

$$NICD_{up} \cong NICD_0 - (N_e + N_s)\left(k_p' - 1\right) \tag{18}$$

$$NICD_p \cong (N_e + N_s)\frac{k_p'}{\Gamma_p'} \tag{19}$$

The total concentration of NICD is then:

$$NICD_{tot} \cong NICD_0 - (N_e + N_s)k_p'\left(1 - \frac{1}{\Gamma_p'} - \frac{1}{k_p'}\right) \tag{20}$$

We now assume the phosphorylation rate of $NICD_{up,b}$ is much faster than the degradation rate of $NICD_{up,ub}$, that is: $k_p \gg \Gamma_{up}$. Under this assumption equation (20) becomes:

$$NICD_{tot} \cong NICD_0 - (N_e + N_s)k_p'\left(1 - \frac{1}{\Gamma_p'}\right) \tag{21}$$

This analysis predicts that the slope in the linear regime is

$$slope_{wt} = -k_p'\left(1 - \frac{1}{\Gamma_p'}\right) \tag{22}$$

For mutant *skd* heterozygotes ($skd^{+/-}$), we expect the phosphorylation rate to change to $k_{p,het}'$. The expression for the total NICD in $skd^{+/-}$ is then:

$$NICD_{tot}^{het} \cong NICD_0 - (N_e + N_s)k_{p,het}'\left(1 - \frac{1}{\Gamma_p'}\right) \tag{23}$$

The ratio of the slopes between the wild type and $skd^{+/-}$ will simply be:

$$\frac{slope_{het}}{slope_{wt}} = \frac{k_{p,het}'}{k_p'} \tag{24}$$

If *skd* is a limiting factor for the formation of the Cdk8 Mediator submodule, it is expected that reducing its copy number from 2 to 1 in $skd^{+/-}$ would result in halving the Cdk8 phosphorylation activity, that is that $k_{p,het}' = \frac{1}{2}k_p'$ of the wild type.

The difference between equations 21 and 23 at $N_s = 0$ gives an expression for $N_e$:

$$N_e = \frac{NICD_{tot}(N_s = 0) - NICD_{tot}^{het}(N_s = 0)}{\left(k_{p,het}' - k_p'\right)\left(1 - \frac{1}{\Gamma_p'}\right)} \tag{25}$$

To check the ratio between wildtype and *skd* het slopes and to estimate $N_e$, we performed linear regression on the data for the mean values of *6S-GFP* expression in *Figure 4h* using the first 3 points of wildtype data (the fourth point is in the saturated regime) and the 4 points of $skd^{+/-}$ data. We note that the data is normalized to the mean fluorescence level of *6S-GFP* expression at $N_s = 0$, so *Equations 21 and 23* are normalized by $NICD_{tot}(N_s = 0)$. This normalization factor does not affect the expressions in *Equations 24 and 25* as it cancels out. The errors are estimated using standard error calculation on multivariate expression (*Clifford, 1973*).

## Estimation of $k_p$

The slope of the normalized linear fit is given by

$$slope_{wt}^{norm} = \frac{slope_{wt}}{NICD_{tot}(N_s = 0)} = \frac{k_p'\left(1 - \frac{1}{\Gamma_p'}\right)}{NICD_0 - N_e k_p'\left(1 - \frac{1}{\Gamma_p'}\right)} \tag{26}$$

Which leads to the following expression for $k_p$

$$k_p = \frac{NICD_0 \Gamma_{up}}{\left(-\frac{1}{slope_{wt}^{norm}} + N_e\right)\left(1 - \frac{\Gamma_{up}}{\Gamma_p'}\right)} \tag{27}$$

This expression allows estimating $k_p$ for different parameter values. We use the calculated values

of $slope_{wt}^{norm}$ and $N_e$. We estimate the steady state amount of NICD in the nucleus, $NICD_0$, ranges between 100 (below that, the concentration is unlikely to activate multiple Notch targets in the nucleus) and 10,000. The upper limit is based on the fact that bicoid concentration is about 10,000 molecules/nucleus (*Gregor et al., 2007*). Since endogenous NICD concentration is so small that it is notoriously hard to detect it in the nucleus using standard imaging techniques (*Couturier et al., 2012*), we estimate that it is not larger than the typical concentration of Bicoid. The estimated range of unphosphorylated NICD is between $\frac{1}{30}\ min^{-1}$ to $\frac{1}{1000}\ min^{-1}$ corresponding to half-lives of the range of 0.5-16 hours, which fits the typical half-lives of proteins. Note, that the analysis in *Figure 3F* shows a half-life of about $120\ min$ in cell culture. Finally, since we assume that $\frac{\Gamma_{up}}{\Gamma_p} \ll 1$ the exact value of $\Gamma_p$ has only a weak effect on the values of $k_p$. For the calculation we take it to be $\Gamma_p = \frac{1}{8}\ min^{-1}$ which is close to the rate observed in *Drosophila* cell culture (*Housden et al., 2013*).

## Dynamic simulations

To study the dynamics of NICD in the nucleus, we numerically solved the dynamic equations corresponding to *Equations 16 and 17*:

$$\frac{d}{dt}NICD_{up} = NICD_0 - NICD_{up}\ - (N_e + N_s)\frac{NICD_{up}}{k_\alpha + NICD_p + NICD_{up}}\left(k_p' - 1\right) \tag{28}$$

$$\frac{d}{dt}NICD_p = (N_e + N_s)\frac{NICD_{up}}{k_\alpha + NICD_p + NICD_{up}}k_p' - NICD_p\Gamma_p' + (N_e + N_s)\frac{NICD_p}{k_\alpha + NICD_p + NICD_{up}}\Gamma_p' \tag{29}$$

The equations were solved using ODE solver in MATLAB, with initial conditions $NICD_{up}(t=0) = NICD_p(t=0) = 0$. The values of parameters used for the simulations are given in *Supplementary file 1*. For simulating wildtype cells, we assumed $N_s = 0$ and $N_e = 5.4$. For simulating $N^{+/-}$ cells, we assumed $P_{NICD}^{N\ het} = \frac{1}{2}P_{NICD}^{wt}$. For simulating cells with two copies of 6SG, we assumed $N_s = 12$.

## Parameter values

| Figure | Parameter values used |
| --- | --- |
| *Figure 4B* | $NICD_0 = 2000\ \frac{\#}{nuc}$, $\Gamma_{up} = \frac{1}{120}\ min^{-1}$, $\Gamma_p = \frac{1}{8}\ min^{-1}$, $k_p = 1, 0.5, 0.25\ min^{-1}$ |
| *Figure 4H* | $NICD_0 = 2000\ \frac{\#}{nuc}$, $\Gamma_{up} = \frac{1}{120}\ min^{-1}$, $\Gamma_p = \frac{1}{8}\ min^{-1}$, $slope_{wt} = -0.0294$, $slope_{skd} = \frac{1}{2}slope_{wt}$, $N_e = 5.4$, $k_p^{wt} = \frac{NICD_0\Gamma_{up}}{\left(-\frac{1}{Slope_{wt}}+N_e\right)\left(1-\frac{\Gamma_{up}}{\Gamma_p}\right)}$, $k_p^{skd} = \frac{NICD_0\Gamma_{up}}{\left(-\frac{1}{Slope_{skd}}+N_e\right)\left(1-\frac{\Gamma_{up}}{\Gamma_p}\right)}$ |
| *Figure 4I* | $NICD_0 = 100 - 10,000\ \frac{\#}{nuc}$, $\Gamma_{up} = \frac{1}{30} - \frac{1}{1000}\ min^{-1}$, $\Gamma_p = \frac{1}{8}\ min^{-1}$, $slope_{wt} = -0.0294$, $N_e = 5.4$, $k_p^{wt} = \frac{NICD_0\Gamma_{up}}{\left(-\frac{1}{Slope_{wt}}+N_e\right)\left(1-\frac{\Gamma_{up}}{\Gamma_p}\right)}$ |
| *Figure 4K* | $NICD_0^{wt} = 2000\ \frac{\#}{nuc}$, $NICD_0^{N\ het} = 1000\ \frac{\#}{nuc}$, $\Gamma_{up} = \frac{1}{120}\ min^{-1}$, $\Gamma_p = \frac{1}{8}\ min^{-1}$, $slope_{wt} = -0.0294$, $N_e = 5.4$, $k_p^{wt} = \frac{NICD_0\Gamma_{up}}{\left(-\frac{1}{Slope_{wt}}+N_e\right)\left(1-\frac{\Gamma_{up}}{\Gamma_p}\right)}$, $N_s = 12$ (for 2xG6S-LacZ) |
| *Figure 4—figure supplement 2* | For A, same parameters as *Figure 4k*<br>B: $NICD_0^{wt} = 200\ \frac{\#}{nuc}$; C: $NICD_0^{wt} = 20000\ \frac{\#}{nuc}$<br>D: $\Gamma_{up} = \frac{1}{30}\ min^{-1}$, E: $\Gamma_{up} = \frac{1}{1000}\ min^{-1}$<br>F: $\Gamma_p = \frac{1}{4}\ min^{-1}$, G: $\Gamma_p = \frac{1}{30}\ min^{-1}$ |

## Lead contact and materials availability

All materials used in this study will be made freely available. Further information and requests for resources and reagents should be directed to and will be fulfilled by the Lead Contact, Brian Gebelein (brian.gebelein@cchmc.org).

## Code availability

All simulation codes are available in the GitHub repository at https://github.com/OhadGolan/NICD-concentration-in-the-nucleus-as-by-binding-site-coupled-NICD-degradation.git (*Golan, 2020*; copy archived at https://github.com/elifesciences-publications/NICD-concentration-in-the-nucleus-as-by-binding-site-coupled-NICD-degradation/tree/master).

## Acknowledgements

We thank members of the Gebelein, Sprinzak and Kopan labs for comments on this work. We thank Drs. Stephen Blacklow, Masato Nakafuku, Kenneth Campbell, and Ertugrul Ozbudak for comments on the manuscript and/or project. We thank Daniel Brissette from the Kopan lab for helping to generate MAML-deficient cell lines. The cdk8[K185] and cycC[Y5] alleles were gifts from Dr. Jessica Treisman (New York University). We thank the *Drosophila* stock centers and Developmental Studies Hybridoma Bank (DSHB) for fly stocks and antibody reagents.

## Additional information

### Funding

| Funder | Grant reference number | Author |
| --- | --- | --- |
| National Science Foundation | 1715822 | David Sprinzak<br>Brian Gebelein |
| National Institutes of Health | CA163653 | Raphael Kopan |
| National Institutes of Health | CA178974 | Rhett A Kovall |

The funders had no role in study design, data collection and interpretation, or the decision to submit the work for publication.

### Author contributions

Yi Kuang, Conceptualization, Data curation, Formal analysis, Investigation, Methodology, Writing - original draft, Writing - review and editing; Ohad Golan, Conceptualization, Formal analysis, Methodology; Kristina Preusse, Joseph Salomone, Matthew R Hass, Investigation; Brittany Cain, Collin J Christensen, Ian Campbell, FearGod V Okwubido-Williams, Formal analysis, Investigation; Zhenyu Yuan, Kenneth H Moberg, Resources; Nathanel Eafergan, Formal analysis; Rhett A Kovall, Resources, Investigation; Raphael Kopan, Conceptualization, Writing - original draft, Writing - review and editing; David Sprinzak, Conceptualization, Formal analysis, Writing - original draft, Writing - review and editing; Brian Gebelein, Conceptualization, Formal analysis, Supervision, Funding acquisition, Investigation, Methodology, Writing - original draft, Writing - review and editing

### Author ORCIDs

Yi Kuang ⬤ https://orcid.org/0000-0002-1509-3620
Collin J Christensen ⬤ http://orcid.org/0000-0002-6932-5554
Rhett A Kovall ⬤ http://orcid.org/0000-0003-0520-1613
David Sprinzak ⬤ https://orcid.org/0000-0001-6776-6957
Brian Gebelein ⬤ https://orcid.org/0000-0001-9791-9061

### Decision letter and Author response

Decision letter https://doi.org/10.7554/eLife.53659.sa1
Author response https://doi.org/10.7554/eLife.53659.sa2

## Additional files

### Supplementary files

• Supplementary file 1. Genetic crosses performed to generate the analyzed progeny. All genotypes of *Drosophila* fly lines mated to generate the offspring and data shown in each experimental Figure.

- Transparent reporting form

### Data availability
All data generated or analysed during this study are included in the manuscript and supporting files.

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
