## [Decision Letter]

**Acceptance summary:**

Your work details a novel mechanism of Notch haploinsufficient phenotypes that is interesting and likely important to a broad range of biological systems given that it is involved in many biological processes.

**Decision letter after peer review:**

Thank you for submitting your article "Enhancer architecture sensitizes cell-specific responses to Notch gene dose via a bind and discard mechanism" for consideration by *eLife*. Your article has been reviewed by Aleksandra Walczak as the Senior Editor, a Reviewing Editor (Hugo Bellen), and two reviewers. The reviewers have opted to remain anonymous.

The reviewers have discussed the reviews with one another and the Reviewing Editor has drafted this decision to help you prepare a revised submission.

Summary:

In this manuscript, the authors have used *Drosophila* genetics, mammalian cell culture and mathematical modeling to show that a specific enhancer configuration can impact Notch signaling in a tissue-specific manner, likely by regulating the level of the Notch intracellular domain. They have found, unexpectedly, that inserting enhancers containing as few as 12 SPS binding sites together with binding sites for a pioneer transcription factor in the genome of Notch[+/+] flies can generate a wing margin phenotype reminiscent of the Notch haploinsufficient wing margin phenotype. This phenotype is quite sensitive to the gene dosages of Notch, a negative regulator of Notch called Hairless, and multiple components of the CDK8 kinase module (CKM) of the Mediator complex. Furthermore, experiments in mammalian cell lines provided evidence that CDK8 plays a key role in regulating Notch1 intracellular domain (N1ICD) turnover. Active transcription is not required for the observed effect in flies and mammalian cells, but formation of the Notch/CSL/Mastermind is. The authors also used quantitative analysis and mathematical modeling to conclude that changes in NICD degradation rate preferentially impact long duration Notch-dependent processes, whereas genetic changes in NICD production rate (consequence of Notch haploinsufficiency) affect both short and long duration processes. Finally, they show that transgenes harboring two copies of an endogenous SPS site behave similarly to the synthetic sites used in the study, and that presence of a pioneer factor binding site is critical for the resulting reduction in Notch signaling in a Notch[+/+] context.

Overall, this is an interesting manuscript with novel and important results. A number of Notch pathway components show significant haploinsufficient phenotypes in animal models and in human patients. However, the basis for tissue-specific phenotypes observed in these diseases and their phenotypic variability is not fully understood. The current work provides a new potential mechanism that might contribute to some aspect of phenotypic variability in these diseases.

Essential revisions:

Some sections of the manuscript are not well explained, and some conclusions do not seem to be fully supported by the data. Moreover, although the idea that duration of signaling determines which cell types are sensitive to SPS sites is appealing, it is only one of the possibilities.

1) Please describe the EMSA data shown in Figure 1 and Figure 1—figure supplement 1 and provide your conclusion. Currently, the EMSA data is not discussed in the Results section. The cooperative binding to SPS seems to be clear for the mammalian complex but not as much for the fly complex. Is it known whether the complex binds the 1x SPS better/stronger that 2x CSL? In other words, have people done competition assays with cold 2x CSL probe in the 1x SPS EMSA and vice versa? Experiments along this line or referring to literature about relative affinity of the RBPJ/Notch/Mam complex for tandem single versus SPS sites might provide clues for the observed difference between single and SPS sites.

2) Subsection “Enhancers with specific TF binding sites can induce a tissue-specific *Notch* phenotype”, "The relative order of GBE and SPS did not matter (Figure 1—figure supplement 2E-F,H)": The data shown in this figure do not support this conclusion. Moreover, no statistics is provided. On the other hand, comparison of Figures 1J and 1K supports this conclusion. Are the quantifications shown in Figure 1—figure supplement 2H accurate? (especially the top two genotypes)

3) How specific is Senexin-A for CDK8? It seems that the only direct evidence used to link mammalian NICD and CDK8 in the manuscript is this compound. If Senexin-A has not been shown to be highly specific for CDK8, knock-down or knockout of CDK8 in a mammalian cell line should be considered to make the link stronger. This is important for two reasons: (a) No direct evidence has been provided for decreased stability of the fly NICD in animals harboring 6SG transgenes; (b) Removing one copy of Cdk8 did not significantly change the wing margin phenotype of *N^+/-^* animals, unlike other components of the CKM complex.

4) In subsection “Cdk8 induces Notch turnover independent from transcription activation” the authors state that "…we found that NICD was stabilized in both mammalian OT11 cells deficient for RPBJ and mK4 cells deficient for three mastermind-like proteins. While the data for the OT11 cells is convincing (Figure 3H), the data for mK4 cells in Figure 3—figure supplement 2 lack a control comparison. In subsection “Cdk8 induces Notch turnover independent from transcription activation”, the authors suggest that the upper band represents NICD with a posttranslational modification. Can the authors show that there is decreased ubiquitination of NICD in Rbpj knockout cells or in cells treated with Senexin-A? Alternatively, since phosphorylation is a key aspect of the mathematical modeling in the manuscript, it might be worth testing whether treating the cell lysate from control cells with a phosphatase increases the mobility of NICD in the gel. Also, please mention what the open and filled arrowheads represent in Figure 3F (perhaps in the legend).

5) In Figure 4J, in the *H^1/+^* genetic background, the removal of one copy of cycC seems to significantly decrease macrochaetae (at p = 8.7X10-3), yet the authors make no mention of this data, stating only that "removing and allele of *skd* or *cdk8* did not significantly alter macrochaetae formation". Although the data supports their conclusion that macrochaetae are relatively unaffected by these genetic manipulations, one wonders if there is any biological significance to the cycC data in this context?

6) Subsection “Altered NICD degradation sensitizes tissues requiring long duration signals”, "We assume that N heterozygotes lower NICD production by one half without impacting NICD.": This assumption is likely to be too simplistic and cell type dependent. Given the importance of *cis*-inhibitory interactions in the Notch haploinsufficient phenotypes in the wing, depending on the relative ligand-to-Notch levels in each context, this number might be lower or higher than one half. In fact, the reason Notch haploinsufficient phenotypes are sensitive to the presence of SPS in some contexts but not others might be the relative contribution of *cis*-inhibition in each context (and/or the relative levels of ligands to Notch in each context). A simplistic example will be to have 10 Notch molecules and 4 ligand molecules in a given cell. Normally, 6 Notch molecules will be competent to generate NICD in this cell. Upon removal of one copy of Notch, only 1 Notch molecule might be available to generate NICD in this cell.

7) Discussion section, paragraphs 1-2: Is there an estimate for the number of SPS sites in the fly genome? The authors mention that Severson et al., suggested that SPS sites contribute to the regulation of about a third of direct Notch1 targets in T-cells. If there are so many SPS sites scattered in the genome, why adding 12 SPS sites can have such a dramatic effect? Does this suggest that the endogenous SPS sites are rarely close to the binding sites of pioneer factors? The mathematical modeling predicted that the contribution of endogenous SPS sites is equivalent to ~5 active synthetic SPS sites. It is worth adding some sentences to the Discussion section to better explain the significance of their findings in the regulation of endogenous Notch signaling, in a wild-type background and in contexts of haploinsufficiency for Notch or its ligands.

8) Discussion section, paragraph 3: In my opinion, linking Notch pathway duration to the observed differences is rather speculative. Although wing margin development takes longer than lateral inhibition, it is not a simple sending-receiving situation that is extended for two days. The direction of signaling and the identity and level of ligands used during wing margin development evolve over time. Moreover, there are other differences between wing margin formation and macrochaetae lateral inhibition. For example, both Delta and Serrate play key and non-redundant roles in wing margin formation, and Serrate loss-of-function phenotype is more severe in the wing margin than the Delta loss-of-function phenotype (for example, Lee et al., 2017). However, loss of Serrate by itself does not have any macrochaetae phenotype, and Serrate only contributes redundantly to macrochaetae lateral inhibition (Pistouli and Delidakis, 2005). Similarly, Serrate only plays a redundant role during wing vein development (Zeng et al., 1998), and based on Figure 4H, even 4 copies of 6SG do not recapitulate the wing vein expansion phenotype of Notch[+/-] animals. The authors should consider modifying this paragraph to acknowledge that Notch signaling duration is just one possibility.

9) Subsection “Cdk8 induces Notch turnover independent from transcription activation”, about the conclusion that removing an allele of each gene of the Cdk8-kinase module significantly suppressed the penetrance and severity of the 6SG-induced wing nicking: Although the data shown in Figure 3 support this conclusion, there is a dramatic variation in the penetrance of wing nicking phenotypes caused by 2 copies of 6SG-lacZ among the figures (compare 1J/1K, 1-S2H, and 2F with 3A). The integration sites used in this study seem to be on a *yw* background. Was the X chromosome crossed out and replaced with a different genetic background when establishing the stocks?

10) Figure 4D-G: Although a subset cells show loss of Cut-positive cells, there seems to be a broadening of Cut expression across the dorso-ventral boundary in all three genotypes with (G6S)n-lacZ. Are these images full projection views? Do the authors have an explanation for this observation (assuming the images are representative for the Cut expression in these genotypes)?

11) Subsection “Quantitative analysis of Enhancer binding site induced Notch turnover”, "We found that GFP levels decreased as a function of added GBE-SPS sites": Potentially related to the above note on Figure 4D-G, although addition of two or three cassettes of G6S clearly results in decreased GFP expression from S-GFP, it appears that the GFP expression domain is broader, with somewhat brighter signal upon addition of one G6S site. Please make sure representative images are shown that match the quantification in 4H. Whether the wing margin region becomes broader in all of these transgenes or not remains an open question. It is worth noting that decreasing CKM gene dosage also affected macrochaetae numbers in the opposite direction of what was expected. Is any other component of the Notch pathway known to be regulated by the CKM complex? For example, if Delta is also a target, a change in the level of Delta might explain both phenotypes (trans-Delta in the case of macrochaetae, cis-Delta in the case of wing margin).

12) The mathematical model used to describe the dynamics of NICD in the cells assumes that dephosphorylation of NICD is absent. While there is no evidence that dephosphorylation of NICD occurs and even if it does there would either be little effect on the model (if dephosphorylation rates were low) or add an additional complexity unnecessary to the model and beyond the scope of the present study, it is an inherent assumption of the model and should be stated as such.

---

## [Author Response]

Essential revisions:Some sections of the manuscript are not well explained, and some conclusions do not seem to be fully supported by the data. Moreover, although the idea that duration of signaling determines which cell types are sensitive to SPS sites is appealing, it is only one of the possibilities.

We revised the text throughout the manuscript to better explain our experimental results and conclusions with each change highlighted under the relevant specific points raised below. Moreover, we acknowledge that while our data is consistent with the duration model, other factors may contribute to the different sensitivities observed in the wing versus macrochaetae tissues. Hence, we added several sentences in the Discussion section highlighting that additional tissue-specific factors besides signal duration could contribute to the differences in phenotypes observed between the wing and macrochaetae.

1) Please describe the EMSA data shown in Figure 1 and Figure 1—figure supplement 1 and provide your conclusion. Currently, the EMSA data is not discussed in the Results section. The cooperative binding to SPS seems to be clear for the mammalian complex but not as much for the fly complex. Is it known whether the complex binds the 1x SPS better/stronger that 2x CSL? In other words, have people done competition assays with cold 2x CSL probe in the 1x SPS EMSA and vice versa? Experiments along this line or referring to literature about relative affinity of the RBPJ/Notch/Mam complex for tandem single versus SPS sites might provide clues for the observed difference between single and SPS sites.

We apologize for this omission. We added several sentences to the Results section to describe the EMSA results and conclusions derived from this line of experimentation (see subsection “Enhancers with specific TF binding sites can induce a tissue-specific *Notch* phenotype” of the revised submission).

To address the question raised about whether the 1xSPS probe is bound better/stronger than the 2xCSL probe by NCM complexes, we performed new EMSA experiments in which the 1xSPS and 2xCSL probes were differentially labeled and added to the same binding reaction with either the mammalian or fly proteins (Figure 1B-C). Hence, direct comparisons can be made in regard to the binding characteristics of the Su(H)/RBPJ transcription factors and the NICD/MAM co-activators to these two probes. These data show that in the absence of co-activators, the 2xCSL probe is bound slightly better by the Su(H)/RBPJ transcription factors than the 1xSPS probe. However, in the presence of NICD/Mam, the 1xSPS is preferentially bound by both the fly and mammalian proteins over the 2xCSL. We describe these results in the modified Results section and included the fly protein EMSAs in Figure 1C rather than in the figure supplement.

We also agree that the mouse proteins appear to be more cooperative than the fly proteins in our EMSAs and we describe and show this observation in Figure 1—figure supplement 1. We decided not to include it in the main text for two reasons: First, our main goal was to test if the fly proteins, which to our knowledge were never tested on SPS sites by EMSAs, generally behave like the mouse proteins. Our data clearly show that the fly proteins do indeed bind cooperatively to the 1xSPS site, but not to the 2xCSL sites. Second, while the main driver of NICD-induced cooperativity is a dimer interface in the Notch ANK domains, which is found in both species (see the data in https://doi.org/10.1101/2020.02.13.948224) slight differences in the polypeptides used in EMSA may impact the degree of cooperativity. Moreover, these differences may reflect the challenge in purification of the NICD and Mam proteins. Thus, we cannot be certain that the differences in cooperativity observed between the mouse and fly proteins in this assay are biologically meaningful.

2) Subsection “Enhancers with specific TF binding sites can induce a tissue-specific Notch phenotype”, "The relative order of GBE and SPS did not matter (Figure 1—figure supplement 2E-F,H)": The data shown in this figure do not support this conclusion. Moreover, no statistics is provided. On the other hand, comparison of 1J and 1K supports this conclusion. Are the quantifications shown in Figure 1—figure supplement 2H accurate? (especially the top two genotypes)

We agree with the reviewers that we overstated this conclusion and reworded the sentence as follows: “i) The *6SG-lacZ* caused notched wings when inserted in another locus and regardless of the order of GBE and SPS, although with differences in penetrance and severity” (subsection “Enhancers with specific TF binding sites can induce a tissue-specific *Notch* phenotype”). We also added statistical tests to Figure 1—figure supplement 2.

The reviewers are also correct that differences in penetrance and severity are sometimes observed between experiments and these differences arise from two sources: First, some differences in penetrance and severity can be attributed to the insertion location: flies with (2 copies in 51C) *vs* (2 copies in 86Fb) *vs* (1 copy in 51C and 1 copy in 86Fb) vary in penetrance/severity. These data are shown and described in Figure 1—figure supplement 2 and statistical tests are now included in this figure to account for the differences in penetrance/severity. Second, we found that the severity/penetrance can vary depending upon environmental changes (i.e. food quality and overcrowding). To deal with this source of variability, all experiments assessing wing notching within each Figure panel were performed with appropriate controls using food made from the same batch and similar fly densities to avoid environmental confounders. We included this information in the Materials and methods section of the paper to better explain our experimental protocol.

3) How specific is Senexin-A for CDK8? It seems that the only direct evidence used to link mammalian NICD and CDK8 in the manuscript is this compound. If Senexin-A has not been shown to be highly specific for CDK8, knock-down or knockout of CDK8 in a mammalian cell line should be considered to make the link stronger. This is important for two reasons: (a) No direct evidence has been provided for decreased stability of the fly NICD in animals harboring 6SG transgenes; (b) Removing one copy of Cdk8 did not significantly change the wing margin phenotype of N^+/-^ animals, unlike other components of the CKM complex.

Senexin A was first described as an optimized inhibitor for CDK8/CDK19 paralogs and the authors claimed that it showed “striking selectivity” (Porter et al., 2012). Senexin A has since been used by multiple other research groups (Galbraith et al., 2017; Wang et el., 2017; Song el al., 2019). To further address this concern, we performed the same half-life measurement with SEL120-34A, another CDK8/CDK19 inhibitor, which was shown to be highly selective and efficient for CDK8/CDK19 (Rzymski et al., 2017). Since Senexin A and SEL120-34A are structurally distinct, they are unlikely to share the same targets other than CDK8/CDK19. Importantly, we observed a similar result with SEL120-34A as to that we observed with Senexin A – inhibitor treatment prolonged NICD half-life. These new findings are described in the Results section and the data are now included in Figure 3—figure supplement 2B and 2D.

For the finding that removing one copy of Cdk8 did not significantly change the wing margin phenotype of *N^+/-^* animals, we would like to point out that catalytic components of multi-subunit enzymes are often not rate limiting factors. In fact, a previous research study found that enzymes, in general, were depleted from the list of haploinsufficiency genes, whereas structural/regulatory proteins that form macromolecular complexes were enriched in haploinsufficiency genes (Kondrashov, et al., 2004). Hence, it is not surprising that CDK8 is not the rate limiting factor in this genetic assay. Instead, the wing phenotype is far more sensitive to the gene dose of the structural/regulatory proteins encoded by the *med12 (kto), med13 (skd*) and the *cycC* genes, which collectively form the CDK8-kinase module with Cdk8. Moreover, this finding is in agreement with prior reports that cycC (CCNC) has been reported to be a haploinsufficient tumor suppressor (Li et al., 2014). We make these points more clearly in the revised Results section and Discussion section.

4) In subsection “Cdk8 induces Notch turnover independent from transcription activation” the authors state that "…we found that NICD was stabilized in both mammalian OT11 cells deficient for RPBJ and mK4 cells deficient for three mastermind-like proteins. While the data for the OT11 cells is convincing (Figure 3H), the data for mK4 cells in Figure 3—figure supplement 2 lack a control comparison. In subsection “Cdk8 induces Notch turnover independent from transcription activation”, the authors suggest that the upper band represents NICD with a posttranslational modification. Can the authors show that there is decreased ubiquitination of NICD in Rbpj knockout cells or in cells treated with Senexin-A? Alternatively, since phosphorylation is a key aspect of the mathematical modeling in the manuscript, it might be worth testing whether treating the cell lysate from control cells with a phosphatase increases the mobility of NICD in the gel. Also, please mention what the open and filled arrowheads represent in Figure 3F (perhaps in the legend).

To address these concerns, we did the following: First, we added the control Western blot for the MAML knockout cells (Figure 3—figure supplement 2E). Second, we performed a Western blot using mK4 cells treated with SEL120-34A and found that this CDK8/CDK19 inhibitor resulted in both increased NICD stability and mobility, consistent with reduced NICD phosphorylation (Figure 3—figure supplement 2D). Third, we treated the protein lysates from mK4 cells with Calf intestinal phosphatase (CIP) and confirmed that the slower migrating (upper) band of NICD is phosphorylation-dependent (Figure 3—figure supplement 2F). Fourth, we added the following description for the arrows in the figure legends of Figure 3 and Figure 3—figure supplement 2: “Open arrow denotes post-translationally modified NICD and closed arrow denotes unmodified NICD.”

5) In Figure 4J, in the H^1/+^ genetic background, the removal of one copy of cycC seems to significantly decrease macrochaetae (at p = 8.7X10-3), yet the authors make no mention of this data, stating only that "removing and allele of skd or cdk8 did not significantly alter macrochaetae formation". Although the data supports their conclusion that macrochaetae are relatively unaffected by these genetic manipulations, one wonders if there is any biological significance to the cycC data in this context?

To address this concern, we provided a more thorough description of the macrochaetae results generated in conjunction with the compound heterozygotes containing either a *Notch* or *Hairless* allele with mutations in the CDK8 kinase module (*cdk8, cycC, kto* (MED12), and *skd* (MED13)). In particular, we added the following statements in the Results section:

“However, we did find that removing an allele of cycC had a small, but significant impact on macrochaetae formation in H;cycC compound heterozygotes. […] Nevertheless, these data suggest that macrochaetae development in general is relatively insensitive to changes in CKM gene dose in comparison to Notch dependent wing margin phenotypes.”

In addition, we also changed the Discussion section to include that the CDK8 kinase module is likely to have additional targets outside of the Notch signaling pathway. Hence, we cannot exclude that additional tissue-specific CDK8 module targets may contribute to the observed phenotypes.

6) Subsection “Altered NICD degradation sensitizes tissues requiring long duration signals”, "We assume that N heterozygotes lower NICD production by one half without impacting NICD.": This assumption is likely to be too simplistic and cell type dependent. Given the importance of cis-inhibitory interactions in the Notch haploinsufficient phenotypes in the wing, depending on the relative ligand-to-Notch levels in each context, this number might be lower or higher than one half. In fact, the reason Notch haploinsufficient phenotypes are sensitive to the presence of SPS in some contexts but not others might be the relative contribution of cis-inhibition in each context (and/or the relative levels of ligands to Notch in each context). A simplistic example will be to have 10 Notch molecules and 4 ligand molecules in a given cell. Normally, 6 Notch molecules will be competent to generate NICD in this cell. Upon removal of one copy of Notch, only 1 Notch molecule might be available to generate NICD in this cell.

As pointed out by the reviewer, we have indeed used the simplified assumption that NICD production is halved for *Notch* heterozygous animals. The main point was to show that the reduction in production affects both long duration and short duration processes. As the reviewer pointed out, *cis*-inhibition is expected to reduce NICD levels in the *Notch* heterozygotes even more, and if such a mechanism impacts one tissue to a higher degree than another, it would potentially make that tissue more sensitive to changes in *Notch* gene dose. We now added several sentences addressing the potential effect of *cis*-inhibition, and how it, and other potential tissue-specific factors, may impact the underlying parameters that contribute to the duration model (Discussion section).

7) Discussion section, paragraphs 1-2: Is there an estimate for the number of SPS sites in the fly genome? The authors mention that Severson et al., suggested that SPS sites contribute to the regulation of about a third of direct Notch1 targets in T-cells. If there are so many SPS sites scattered in the genome, why adding 12 SPS sites can have such a dramatic effect? Does this suggest that the endogenous SPS sites are rarely close to the binding sites of pioneer factors? The mathematical modeling predicted that the contribution of endogenous SPS sites is equivalent to ~5 active synthetic SPS sites. It is worth adding some sentences to the Discussion section to better explain the significance of their findings in the regulation of endogenous Notch signaling, in a wild-type background and in contexts of haploinsufficiency for Notch or its ligands.

To address this concern, we have added a paragraph in the Discussion section describing the current literature in regards to numbers of SPS sites in the genome. In this paragraph, we discuss that while the number of functional SPS sites in the *Drosophila* genome has not been accurately determined, we do have estimates from mammalian cells. Moreover, we also have some estimates for the number of Su(H) binding loci in a *Drosophila* wing cell line, although this study didn’t differentiate CSL vs. SPS sites. In addition to incorporating this information into the discussion, we also discussed the possibility that concentrating the SPS sites at a specific locus may have an “avidity” impact such that any NICD molecule recruited to the locus has a very high probability of being marked for degradation. By integrating these data into the manuscript, we believe it puts our finding that transgenes with as few as 12 SPS sites can induce phenotypes into a better context relative to the entire endogenous genome.

8) Discussion section, paragraph 3: In my opinion, linking Notch pathway duration to the observed differences is rather speculative. Although wing margin development takes longer than lateral inhibition, it is not a simple sending-receiving situation that is extended for two days. The direction of signaling and the identity and level of ligands used during wing margin development evolve over time. Moreover, there are other differences between wing margin formation and macrochaetae lateral inhibition. For example, both Delta and Serrate play key and non-redundant roles in wing margin formation, and Serrate loss-of-function phenotype is more severe in the wing margin than the Delta loss-of-function phenotype (for example, Lee et al., 2017). However, loss of Serrate by itself does not have any macrochaetae phenotype, and Serrate only contributes redundantly to macrochaetae lateral inhibition (Pistouli and Delidakis, 2005). Similarly, Serrate only plays a redundant role during wing vein development (Zeng et al., 1998), and based on Figure 4H, even 4 copies of 6SG do not recapitulate the wing vein expansion phenotype of Notch[+/-] animals. The authors should consider modifying this paragraph to acknowledge that Notch signaling duration is just one possibility.

We agree that the proposed duration model is correlative by nature and it is difficult to definitively test the model (i.e. alter signal duration) in the wing margin and macrochaetae. Hence, we expanded the Discussion section to point out that while our data is consistent with the duration model, other tissue-specific factors and parameters may also contribute to the observed differences in sensitivity between the wing margin and macrochaetae.

9) Subsection “Cdk8 induces Notch turnover independent from transcription activation”, about the conclusion that removing an allele of each gene of the Cdk8-kinase module significantly suppressed the penetrance and severity of the 6SG-induced wing nicking: Although the data shown in Figure 3 support this conclusion, there is a dramatic variation in the penetrance of wing nicking phenotypes caused by 2 copies of 6SG-lacZ among the figures (compare 1J/1K, 1-S2H, and 2F with 3A). The integration sites used in this study seem to be on a yw background. Was the X chromosome crossed out and replaced with a different genetic background when establishing the stocks?

As for the variability between experiments, please see our response to point #2 above regarding two sources of phenotype variability. First, while the *6SG-lacZ* transgenes induce wing notching phenotypes regardless of chromosomal integration site, the insertion location can impact the penetrance and severity of the phenotype. Thus, the data in Figure 2F (*6SG-lacZ-51C/6SG-lacZ-51C*) and Figure 3A (*6SG-lacZ-51C/+; 6SG-lacZ-86Fb/+*) are not comparable since the two alleles of *6SG-lacZ* are on different chromosomes, and we found that flies with *6SG-lacZ-51C/+; 6SG-lacZ-86Fb/+* have a higher wing-phenotype penetrance than flies with either *6SG-lacZ-51C/6SG-lacZ-51C* or *6SG-lacZ-86Fb/6SG-lacZ-86Fb* (Figure 1—figure supplement 2H). Second, food quality and fly density can impact the penetrance and severity of the wing nicking phenotype observed, and we described both this result and how we addressed it in point #2 above. As for the X chromosome, all of our transgenes, including the GBE-CSL reporters, the GBE alone reporters, and the SPS alone reporters that do not induce phenotypes, were created and tested in an identical manner as the transgenes containing the SPS sites. Moreover, we always used *yw* as our control, even when crossing with the *Hairless* and *Notch* heterozygous animals. Given that the *yw* X chromosome never showed phenotypes on its own, nor with any of the other reporters except those with GBE-SPS or when crossed to *Notch* heterozygous animals, we think that it is highly unlikely to contribute in any meaningful way to the observed phenotypes.

10) Figure 4D-G: Although a subset cells show loss of Cut-positive cells, there seems to be a broadening of Cut expression across the dorso-ventral boundary in all three genotypes with (G6S)n-lacZ. Are these images full projection views? Do the authors have an explanation for this observation (assuming the images are representative for the Cut expression in these genotypes)?

The images in Figure 4D-G were taken as Z-stacks before being max-projected. To address this concern, we carefully analyzed all of the wing disc data and found that while the width of the Cut-expressing cells does vary from wing disc to wing disc, we observe similar variation in all genotypes. Also, see our response to the related point 11 below.

11) Subsection “Quantitative analysis of Enhancer binding site induced Notch turnover”, "We found that GFP levels decreased as a function of added GBE-SPS sites": Potentially related to the above note on Figure 4D-G, although addition of two or three cassettes of G6S clearly results in decreased GFP expression from S-GFP, it appears that the GFP expression domain is broader, with somewhat brighter signal upon addition of one G6S site. Please make sure representative images are shown that match the quantification in 4H. Whether the wing margin region becomes broader in all of these transgenes or not remains an open question. It is worth noting that decreasing CKM gene dosage also affected macrochaetae numbers in the opposite direction of what was expected. Is any other component of the Notch pathway known to be regulated by the CKM complex? For example, if Delta is also a target, a change in the level of Delta might explain both phenotypes (trans-Delta in the case of macrochaetae, cis-Delta in the case of wing margin).

To address the concern regarding the GFP levels and pattern in the wing discs shown in Figure 4D-G, we reanalyzed the data and selected new images that are representative of GFP levels, pattern, and the width of the Cut expression domain. While looking at all of the discs analyzed in this experiment, we did notice variation in the width of Cut expression domain between imaginal discs. However, besides observing gaps in the Cut expression domain for those flies carrying 6, 12, or 18 *GBE-SPS* sites (marked by arrowheads in Figure 4E-G), the variation was similar between genotypes. Moreover, it is important to point out that all of the data collected from each imaginal disc was included in the quantified data shown in Figure 4H, as each dot on the graph is the average GFP intensity in the Cut-positive cells from a single wing imaginal disc and none of the discs were excluded from this analysis.

The reviewer also raised a second point: Is any other component of the Notch pathway known to be regulated by the CKM complex? To our knowledge, CKM activity is restricted to the nucleus and does not modify the Delta ligand or the unprocessed Notch receptor. However, CKM activity is known to phosphorylate other transcriptional regulators including Mastermind, CycC, Med12, and Med13. Taking these potential targets into account, removing an allele of CKM components would still be predicted to increase Notch signal strength, which is consistent with our current model. In addition, to these points, we also included a statement that we cannot exclude the possibility that CKM activity alters additional transcriptional regulatory proteins, and that they could contribute to the observed phenotypes. We make these points clearer in the modified Discussion section.

12) The mathematical model used to describe the dynamics of NICD in the cells assumes that dephosphorylation of NICD is absent. While there is no evidence that dephosphorylation of NICD occurs and even if it does there would either be little effect on the model (if dephosphorylation rates were low) or add an additional complexity unnecessary to the model and beyond the scope of the present study, it is an inherent assumption of the model and should be stated as such.

We agree with the reviewer that adding additional complexity to the model is beyond the scope of the present study. We have specified in the new version that we assume no dephosphorylation of NICD is considered in the model in the Results section.